# Essential-Oil-Loaded Nanoemulsion Lipidic-Phase Optimization and Modeling by Response Surface Methodology (RSM): Enhancement of Their Antimicrobial Potential and Bioavailability in Nanoscale Food Delivery System

**DOI:** 10.3390/foods10123149

**Published:** 2021-12-20

**Authors:** Sana Yakoubi, Isao Kobayashi, Kunihiko Uemura, Mitsutoshi Nakajima, Hiroko Isoda, Riadh Ksouri, Moufida Saidani-Tounsi, Marcos A. Neves

**Affiliations:** 1Graduate School of Life and Environmental Sciences, University of Tsukuba, Tennoudai 1-1-1, Tsukuba 305-8572, Japan; nakajima.m.fu@u.tsukuba.ac.jp (M.N.); isoda.hiroko.ga@u.tsukuba.ac.jp (H.I.); 2Laboratory of Aromatic and Medicinal Plants, Center of Biotechnology, Technopark of Borj-Cedria (CBBC), BP 901, Hammam-Lif 2050, Tunisia; ksouririadh@gmail.com (R.K.); tounsi.moufidaa@gmail.com (M.S.-T.); 3Alliance for Research on the Mediterranean North Africa (ARENA), University of Tsukuba, Tennoudai 1-1-1, Tsukuba 305-8572, Ibaraki, Japan; isaok@affrc.go.jp; 4Faculty of Science of Tunis, Department of Biology, University of Tunis El Manar, Tunis 2092, Tunisia; 5Division of Food Engineering, Food Research Institute, NARO, Kannondai 2-1-12, Tsukuba 305-8517, Ibaraki, Japan; uemura@affrc.go.jp

**Keywords:** essential-oil-loaded nanoemulsion, RSM, droplet size, interfacial tension, antimicrobial potential, electrostatic properties, colloidal system, nanoemulsion stabilization mechanism

## Abstract

Nanoencapsulation is an attractive technique used for incorporating essential oils in foods. Thus, our main goal was to formulate a novel nanoemulsion (NE) with nanoscale droplet size and lowest interfacial tension in the oil–water interface, contributing positively to the stability and the enhancement of essential oil potential. Thereby, response surface methodology (RSM), with mixture design was used to optimize the composition of the NE lipid phase. The essential oil combinations were encapsulated through high-pressure homogenization (HPH) with the binary emulsifier system (Tween 80: Gum Arabic). Then, the electrophoretic and physical properties were evaluated. We also conducted a follow-up stability and antimicrobial study that examined the stabilization mechanism of optimal NE. Thereafter, the effect of nanoencapsulation on the essential oil composition was assessed. The RSM results were best fitted into polynomial models with regression coefficient values of more than 0.95. The optimal NE showed a nanometer-sized droplet (270 nm) and lower interfacial tension (~11 mN/m), favoring negative ζ-potential (−15 mV), showing good stability under different conditions—it synergistically enhances the antimicrobial potential. GC-MS analysis showed that the use of HPH affected the active compounds, consistent with the differences in linalool and 2-Caren-10-al content. Hence, the novel nanometric delivery system contributes to food industry fortification.

## 1. Introduction

In recent decades, consumers have preferred eco-friendly products without synthetic chemicals due to their negative secondary effects. That is why the use of natural aromatic compounds and flavors in different industry products is significant [1,2]. Thus, food technology has emerged to produce and deliver biomolecules and essential oils (EO) with a variety of applications in modern life. Among them, the cumin-, carvi-, and coriander-EOs, are thought to be potential natural food preservatives and antimicrobial agents [1,2,3]. It was reported that terpenoids were an important component of essential oils [1,3]. Usually, the essential oil potential could be attributed to the presence of its major compounds.

In nanoemulsion systems (NE), properties such as stability, rheology, appearance, color, and texture depend on the particle size distribution as well as the nature of the essential oil and its relative chemical composition [1,3]; therefore, it is worth highlighting the interesting impact of the synergy between EOs and the delivery system nowadays. Consequently, their valorization would be achievable by optimizing the EOs composition in NE. In addition, the ultimate goal of optimization of the nanoemulsion aqueous and/or organic phase is often to achieve minimum droplet size and maximum stability with a good potential [3]. Thus, the encapsulation of the three EOs presents an important issue of interest for the food-based product, which could improve their limited application in the food industry due to their low solubility and high sensorial properties (e.g., odor).

In the food industry, nanoemulsions are typically produced using high-energy emulsification methods, such as high-frequency ultrasound (HFU) [2]. However, the study of Yakoubi et al. [2] showed that this technique was unable to generate essential-oil-loaded nanoemulsion in the nanometric size. Another technique commonly used is high-pressure homogenization (HPH) [4]. Due to its consumption of sufficient energy and surfactant, it provides droplets with smaller size, lower polydispersity, and higher stability than other techniques [4]. In addition, the type and dose of the compounds in the nanoemulsions play a significant role in the determination of their characteristics. Thus, in order to improve the physical, rheological, and stability properties of nanoemulsions, the qualitative and quantitative selection of organic and aqueous phases are important factors. Otherwise, based on the findings of Krishnan et al. [5], the emulsifiers’ protective effect was not similar for all the constituents. In this case, they followed an order of volatile compound effect (i.e., *p*-cymene> cuminaldehyde > γ-terpinene). In this respect, they highlight the importance of the composition and the nature of the compounds forming the oil phase in the physicochemical properties, as well as the stability of the nanoemulsion.

Emulsifiers are molecules with a surface activity that reduces interfacial tension due to their adsorption at the interface of oil droplets. The most commonly used emulsifiers in the food industry are small molecules of synthetic surfactants, i.e., polyoxyethylene sorbitan esters of fatty acids, and “natural” emulsifiers, i.e., amphiphilic proteins, phospholipids, and polysaccharides [6,7,8]. Their molecular weight influences the kinetics of adsorption of amphiphilic molecules during the emulsification process. Small molecules such as Tween are more suitable to produce a smaller particle size than caseinate or β-lactoglobulin under similar conditions of homogenization, possibly because they are adsorbed faster on the interface [7]. Tween 80 (polysorbate 80, polyoxyethylene sorbitan monooleate), a non-ionic surfactant widely used as an emulsifier in food products, is endowed by a high surface activity at the oil–water interface due to its low molecular weight [7,9]. Further, the study of Tahir et al., 2020 [8], suggests that the reduced size of encapsulated vitamin D may be due to the use of Tween 80 as a surfactant, which is more effective for reduction in the size of nanoparticles and offers greater stability. Meanwhile, Gum Arabic (GA) has been recognized as an efficient emulsifying agent of the food industry. GA stabilizes the oil-in-water emulsions over a wide range of pH, temperature, and ionic strength. It is considered to be eco-friendly [10]. Moreover, GA is a highly branched, neutral, or slightly acidic polysaccharidic complex, containing about 2% of polypeptide. Furthermore, GA was able to form thick viscoelastic films at the oil–water interface. It should be noted that the use of GA as an emulsifier can considerably improve the stability of emulsions to aggregation by reducing the strength of the Van Der Waals interactions between them [6,11]. The major advantage of GA as an emulsifying agent is that it is a reliable emulsifier as the emulsions stay stable for a very long time. Furthermore, the ability of GA to inhibit the protein auto-assemblage has already been reported [6]. Our previous work conducted by Yakoubi et al. [2] showed that the use of the binary emulsifier mixture (Tween 80: Gum Arabic 0.75: 0.25 *v*/*v*) as surface-active, could effectively stabilize the NE under different conditions and was able to generate nanoscale food delivery system-based tocopherol [2].

In the present study, the cumin, carvi, and coriander EO were encapsulated by the high-pressure homogenizer (HPH) using the binary emulsifier system (Tween 80: Gum Arabic 0.75: 0.25 *v*/*v*). Thus, our main goal in this study was the optimization of the nanoemulsion lipid phase in order to obtain the lowest nanometer-sized droplets, the highest stability in different conditions, and the highest antimicrobial and antioxidant potential of interest for food-based product application. The underlying purposes of this study were (i) to determine the effect of the oil-phase variability on the nanoemulsion properties, (ii) the effect of the high-pressure homogenization and the binary emulsifier system (T80: GA 0.75: 0.25 *v*/*v*) on the biochemical compositions of essential oils, and (iii) improvement of the biomolecules bioavailability in the food-delivery system. The response surface methodology (RSM) was also highlighted in the optimization process of the EO content in the NE dispersed phase.

## 2. Materials and Methods

### 2.1. Materials

Polyoxyethylene (80) sorbitan monooleate (Tween^®^ 80 (T80)) and Gum Arabic (GA) (Sigma-Aldrich S.A., St. Louis, MO, USA) were all of reagent grade and manufactured by Wako Pure Chem. Ind. Ltd. (Tokyo, Japan). Ultrapure Milli-Q water (resistivity 18.2 MΩ·cm; Millipore Corp., Bedford, MA, USA) was used to prepare the aqueous stock solutions for the nanoemulsion preparation. *Bacillus subtilis* subsp. spizizenii JCM 2499 and *Echerichia coli* (JCM 1649) (Japan Collection of Microorganisms) [12] were kindly provided by the National Food Research Institute (NFRI, Tsukuba, Japan). Nutrient agar (product code 05514) and nutrient broth (product code 05511), both manufactured by Nissui Pharm. Co. (Tokyo, Japan), were used as substrates for microbial growth. Sterilized Ringer’s solution (product code 391-01421; NaCl 2.25 g/L, KCl 0.105 g/L, CaCl2·2H2O 0.16 g/L; Nihon Seiyaku Co., Tokyo, Japan) was used as a physiological salt solution for serial dilution of pre-culture media for *B. subtilis* and *E. coli* spores. Seeds of *C. cyminum*, *C. carvi*, *and C. sativum* were collected in June 2017, from the north of Tunisia (Teborba); latitude 36.841754″ (N); longitude 9.861912″ (E) and the altitude is 637 m. The precipitation average was 400–500 mm/year and the monthly average temperature was 30 °C. The spices were crushed and used immediately for the oil extraction. Then their essential oils were used as the disperse phase in the O/W emulsions.

### 2.2. Chromatographic Analysis

The EOs were dissolved in 1 mL dichloromethane in order to prepare them for the GC-MS analysis. The GC-FID apparatus was equipped with an Agilent gas chrοmatοgraph series 7890-A, a flame ionization detectοr (FID) and a silica capillary HP-5 cοlumn (30 m × 0.32 mm i.d.; film thickness 0.25 µm) [13]. 0.2 µL of pure EO was injected into the GC. The injector and detector temperatures were fixed at 250 °C and 280 °C, respectively. The parameters of GC-FID are as follows: Nitrοgen as carrier gas; 1 mL/min as flοw rate; 60–210 °C as οven temperature prοgram at the rate οf 4 °C min, followed by 240 °C at the rate οf 20 °C min and finally, held isοthermally fοr 8.5 min; 1:50 as split ratiο. The GC was cοupled with a 5975-C mass spectrοmeter. Thus, the GC/MS adopts the split mοde (1:50), a 0.1 μL of pure EO was injected intο a capillary cοlumn. Hither the carrier gas and the iοnizatiοn vοltage were Helium and 70 eV, respectively. The ion source and interface temperatures were 230 °C and 280 °C, respectively; likewise, the mass range was set frοm 45 tο 550 amu. The methοd of the standard “n-alkanes” was used for determination of the retentiοn indices fοr all cοmpοunds. The tentative identification of the oil constituents was based on a comparison of their retention indices relative to (C7–C20) n-94 alkanes (Analytical Reagent, LabScan, Ltd., Dublin, Ireland) with those of literature or with 95 authentic compounds available in our laboratory [2].

### 2.3. Emulsification Method

A high-energy method was used for the formulation of oil-in-water nanoemulsion (O/W NE) [2]. For this, two phases were prepared. The lipidic phase is composed of different combinations of the cumin, carvi, and coriander essential oils, fitted by RSM. Meanwhile, the aqueous phase contained the binary emulsifier mixture (T80: GA 0.75:0.25 *v*/*v*). The emulsification process of O/W NE was performed in two steps at room temperature. Firstly, the primary O/W emulsions were passed by high-speed homogenization using Polytron PT-3100, Kinematica, Switzerland, a rotor-stator homogenizer at 10,000 rpm/5 min. Secondly, nanoscale-O/W emulsions were prepared by high-pressure homogenization (HPH) (NanoVater NV200, Yoshida Kikai, Aichi, Japan) at 100 MPa for 4 cycles [4]. The post-treatment characteristics of their physico-chemical and biological properties were analyzed, whilst the other techno-functional properties were analyzed only for the optimal NE [4].

### 2.4. Mean Droplet Size and Size Distribution of O/W Nanoemulsions

The droplet size distribution of the obtained O/W NE was carried out using laser diffraction employing a Laser Diffraction Particle Size Analyzer endowed by a polarization intensity differential scattering technology (PIDS) (LS 13320, Beckman-Coulter, Inc., Brea, CA, USA), measuring the particle size in the range [0.017–2000 µm] [4]. The mean droplet size was determined by the surface-weighted mean diameter, and so-called Sauter mean diameter (*d*_3.2_), as described in Equation (1):(1)$$d_{3.2}=\frac{volume}{surface\;area}=\frac{\sum^{\;}ni\;di\;3\;}{\sum^{\;}ni\;di\;2\;}\;\;$$
where *ni* is the number of droplets and *di* is the droplet diameter. All measurements were performed in triplicate. The refractive indices of cumin EO, carvi EO, and coriander EO were as follows: 1.4730, 1.5003, and 1.462. The calculations for the polydispersity parameters are defined in the ISO standard document 13321:1996 E and ISO 22412:2008.

### 2.5. Effect of Nanoencapsulation on the Antioxidant Activity of EOs

Antioxidant properties of the obtained O/W NE were evaluated by DPPH radical scavenging activity according to the method described elsewhere [14]. A total of 4 mL of different O/W NE dissolved in methanol were separately added to a 1 mL of DPPH radical in methanol solution (0.2 mM). The mixture was shaken vigorously and allowed to stand for 30 min, after which the absorbance of the resulting solution was measured at 517 nm with a spectrophotometer (Shimadzu UV-1240, Kyoto, Japan). BHT was used as a positive control. Inhibition of free radical DPPH as a percentage (I%) was calculated as follows:*I* (%) = 100 × (*A*_blank_ − *A*_sample_)/*A*_blank_
where *A*_blank_ is the absorbance of the control and *A*_sample_ is the absorbance of the O/W NE.

### 2.6. Effect of Nanoencapsulation on the Antimicrobial Activity of EOs

Both Gram-negative and Gram-positive bacteria were used to assess the effect of nanoencapsulation on the antimicrobial activity of EOs and to determine the optimal NE having the higher antimicrobial potential. Thus, *Escherichia coli* (*E. coli*) as Gram-negative, represents the *Enterobacteriaceae*, meanwhile *Bacillus subtilis* (*B. subtilis*), as Gram-positive, represents the *Bacillaceae*. *B. subtilis* and *E. cοli* were expοsed tο the treatments by the blank (phosphate buffer) and by the different emulsions from which the combinations were generated by the software. Serial dilutiοns (frοm 10^9^ tο 10^2^ CFU/mL) were prepared using the physiοlοgical salt sοlutiοn.

#### 2.6.1. Counting Test

The automatic counting test was performed by the digital microscope culturing system (S-12, BiοmaticTM, DCMS, MicrοBiο Cο., Tokyo, Japan), after pouring an agar thin top layer over microbe O/W NE to support colonies’ growth until having at least 10,000 CFU/μL. All the preliminary experiments began with a bacterial concentration of 10^5^ and 10^6^ CFU/mL for *B. subtilis* and *E. coli*, respectively. Thus, 20 mL of nutrient agar were poured onto the empty Petri dishes. Simultaneously, the bacterial spore suspension was melted with O/W NE (*v*/*v*). After 30 min magnetic stirring, 100 μL οf the mixture was spread οn tοp οf the sοlidified agar, additionally, a second thin layer of agar was pοured before the incubation at 37 °C up to 24 h. Finally, the viable cell count was determined by the digital microscope. Two Petri dishes without O/W NE, just with *E. coli* and *B. subtilis* colonies, were considered as a negative control. All experiments were perfοrmed in triplicate at rοοm temperature.

#### 2.6.2. Determination of Minimal Inhibitory Concentration (MIC) and Minimal Bactericidal Concentration (MBC)

The broth dilution method in Mueller Hinton Broth (MHB) was used for the determination of the MIC of the optimal O/W NE [15]. Fresh broth cultures of *E. coli* and *B. subtilis* after 24 h incubation were individually adjusted to 0.5 McFarland in 0.85% (*w*/*v*) SSB (10^8^ CFU/mL for *E. coli* and *B. subtilis*) by using the nephelometer. The O/W NE dissolved in the DMSO solution was filtered using a sterile syringe filter (TPP) with pore size 0.22 µm. A total of 6 different serial dilutions of O/W NE, from 0.5 µg/mL to 10 mg/mL, were prepared in MHB [16]. The *E. cοli* and *B. subtilis* concentrations were adjusted tο 5 × 10^6^ CFU/mL, subsequently centrifuged at 4500× *g*/10 min. Cell pellets (0.5 McFarland) were directly mixed with 1 mL of six different concentrations of O/W NE in MHB in 24-well plates. They were incubated at 37 °C for 24 h. Additionally, the initial bacterial cell number was checked by swabbing on MHA from serial dilutions in SSB. A total of 30 µg/mL of chloramphenicol in MHB and blank consisting of only MHB were used as a positive control and negative control, respectively. MIC test was carried out in terms of OD_600_ values using a microplate reader (FLUOstar Omega, BMG Labtech, Cary, NC, USA). All experiments were repeated three times. MIC refers to the lowest concentration of O/W NE in MHB at which the bacterial growth was inhibited after 24 h of incubation. In order to prepare the colony for enumeration, 100 µL samples from each MIC dilution were spread out on the MHA medium and incubated at 37 °C for 24 h. The minimum dilution of the O/W NE on the MHA that provided growth inhibition was identified as MIC. Whereas the MBC was the lοwest cοncentratiοn that cοuld kill 99.9% οf treated cells *E. cοli* or *B. subtilis*.

### 2.7. Design of Experiments

The Design Expert (13) software was performed to assess the effect of concentrations of three different essential oils, Cumin essential oil (EO_CU_), Carvi essential oil (EO_CA_), and Coriander essential oil (EO_CO_), on four response variables: the emulsion droplet size (*d*_3.2_ nm), the antiradical scavenging DPPH activity (IC_50_ mg/mL) and the viable *B. subtilis,* and *E. coli* colony enumeration (CFU/mL). Thus, the I-optimal design fitted the data and was constrained by this relationship:EO_CU_ + EO_CA_ + EO_CO_ = 100 %,     where     0 < EO < 1

The areas with stable O/W NE were identified by the pseudo-ternary phase diagrams among 17 runs. All O/W NE were formulated and analyzed on the same day without further storage. All the experiments were carried out in triplicate, and all parameters were calculated and used for modeling. Based on the I-optimal approach effect of the input variables (EO_CU_, EO_CA,_ and EO_CO_) on *d*_3.2_, DPPH-IC_50_ and viable microbe cells were evaluated, then the optimal combination was determined by the desirability approach. The fitting response value was determined using the reduced cubic model (Equation (2)) for *d*_3.2_, IC_50,_ and *B. subtilis* colony. The *E. coli* colony was adjusted by the quadratic model (Equation (3)); the mathematical models are given by:(2)$$y=\overset{q}{\underset{i=1}{\sum }}\beta i\;xi+\overset{q-1}{\underset{i<j}{\sum }}\overset{q}{\underset{j}{\sum }}\beta ij\;xixj+\overset{q-1}{\underset{i<j}{\sum }}\overset{q}{\underset{j}{\sum }}\delta ijxixj\;\left(xi-xj\right)$$
(3)$$y\prime =\overset{q}{\underset{i=1}{\sum }}\beta i\;xi+\overset{q-1}{\underset{i<j}{\sum }}\overset{q}{\underset{j}{\sum }}\beta ij\;xixj$$
where *y* is the predicted response variable (*d*_3.2_, IC_50_ and *B. subtilis* colony), *y’* is the response of *E. coli* colony, *i*, *j*, and *k* are the number of ingredients in the mixture, *βi* is the first-order coefficient, *βij* is the second-order coefficient, *x_i_*, *x_j_*, and *x_k_* are the corresponding EO_CU_, EO_CA,_ and EO_CO_, respectively [17]. ANOVA at *p*-value < 0.05 was used to evaluate the statistical significance.

### 2.8. Interfacial Tension Measurement

The pendant drop method was used for the determination of the interfacial tension using an interfacial tensiometer (DM-501, Kyowa Interface Science Co., Ltd., Saitama, Japan). Briefly, Solutions were placed in a 1 mL syringe followed by the formation of one drop manually at the tip of a Teflon coated needle (22 G). Subsequently, interfacial tension was calculated automatically by the analysis software depending on the shape of the drop after reaching the maximum volume before detachment. All fluid measurements were performed at room temperature.

### 2.9. Electrophoretic Properties

The surface charge (ζ-potential) of the optimal O/W NE was carried out at 25 °C using a ζ-potential analyzer (Zetasizer Nano ZS, Malvern Instruments Ltd., Worcestershire, UK). The optimal O/W NE was diluted with a phosphate buffer with appropriate pH (0.5:0.5 *v*/*v*, pH = 7). ζ-potential values were gathered over 30 continuous readings.

### 2.10. Physical Stability under Different Treatments

#### 2.10.1. Droplet Size Determination in Different Ionic Strength and Temperature

The physical stability (*d*_3.2_ nm) of the optimal O/W NE was assessed in different ranges of ionic strength (0–500 mM NaCl) and temperature (20–100 °C) after a centrifugation process at 2400× *g* for 15 min (Universal 32R, Hettich, Salford, UK) to accelerate NE destabilization. For ionic strength treatment, the optimal O/W NE was diluted in a NaCl solution at different concentrations (W/T (without any salt addition), 100, 200, 300, 400, and 500 mM NaCl) (*v*/*v*). For temperature treatment, 8 mL of O/W NE was immersed in a water bath (B-100, Buchi, Flawil, Switzerland) at different temperatures (W/T (without temperature treatment), 20, 40, 60, and 100 °C) for 30 min.

#### 2.10.2. Long-Term Storage Stability of the Optimum NE

The optimal O/W NE was stored at 4 °C in neutral pH without salt addition for 10 months with Sodium azide (0.02% (*w*/*w*)) to prevent microbial growth. The *d*_3.2_ was determined each month. Further, the drop growth of the optimal O/W NE was tested based on Equation (4):(4)$$R=\frac{\left(d_{3.2}\;n-d_{3.2}\;0\right)}{d_{3.2}\;0}$$

*R* is the droplet growth ratio, *d*_3.2_ *n* and *d*_3.2_ 0 is the droplet size diameter of TO-NE at days *n* and 0, respectively.

### 2.11. Statistical Analysis

I–optimal mixture design, referring to different models, was fitted for assessing the effect of the O/W NE lipid-phase composition on their physical and biological properties. The analyses were examined in triplicate. Thus, the statistical analysis, graphical representations, and optimization were carried out by Design Expert 13 software (Stat–Ease Inc., Minneapolis, MN, USA). The results were subjected to ANOVA one-way analysis and Tukey’s test using SAS Ver. 8.2, (Sas Institute Inc., Cary, NC, USA). A 5% as significance level (*p*-value < 0.05) was considered in all experimental data.

## 3. Results

### 3.1. Chromatographic Analysis of Essential Oils

The bulk essential oils were obtained by hydrodistillation. *C. sativum* EO presented a low yield (0.23%) in comparison with these of *C. cyminum* (1.55%) and *C. carvi* (2.71%) growing in the same region. The difference in this value compared to the others was attributed to several factors, such as the chemical composition. The GC–MS analyses of cumin EO revealed the presence of 30 components constituting 99.89% of the total oil. Cuminaldehyde (26.94%), Limonene (26.04%), γ-Terpinene (14.01%), 3-Caren-10-al (11.64%), β-pinene (11.10%), and 2-Caren-10-al (5.98%) were the main cumin oil components, representing 94.32% of the total. The carvi essential oil analysis indicated clearly that carvone (70.36%) and limonene (26.78%) were the main components; while the *Coriandrum* essential oil depicted eighteen volatile compounds and the linalool (68.31%) was the major compound; there has been some agreement with regards to the oil composition. Nevertheless, the results show a difference between the volatile compounds percentage, which could be due to many causes, such as environmental, seasonal, and geographical variations and the extraction process [18,19] (Table 1 and Figure 1). With the exception of the existence of 1-phenyl-1,2-ethanediol, Estonian cumin (EO) [20] had content comparable to our samples. The results of carvi EO varied from early findings. For instance, the two major compounds of Chinese carvi EO were (R)-carvone (51.62%) and D-limonene (38.26%) [21], followed by pinene (5.21%), cis-carveol (5.01%), and myrcene (4.67%) [22], similar to EO extracted from carvi cultivated in Europe and North America [23].

### 3.2. Nanoemulsion Formulation

A serial combination between cumin, carvi, and coriander essential oils was generated by the Design Expert software to encapsulate them by the binary emulsifier mixture (T80: GA 0.75: 0.25, *v*/*v*) (Table 2). Polytron at 10,000 rpm/5 min was used to produce primary emulsions that were then passed by the high-pressure homogenizer (HPH) at 100 MPa to formulate nanoscale emulsions. A response surface methodology design was built in order to optimize the composition of NE lipid phase. Only the four responses (droplet size diameter, antiradical scavenging activity, *B. subtilis,* and *E. coli* colony number) were taken into consideration in the RSM analysis because having a nanoscale diameter suggests a small polydispersity index (PDI). So PDI was excluded from RSM analysis.

### 3.3. Optimization of Essential-Oil-Loaded Oil-in-Water Nanoemulsion Composition by Response Surface Methodology (RSM)

#### 3.3.1. Fitting for the Model

Based on these criteria, low standard deviation (SD), low predicted sum of squares (SS_pred_), and high R-squared (R^2^), furthermore a *p*-value less than 0.05 whereas *p*-value of lack of fit higher than 0.05, the different models of the four responses were determined [24]. Considering the following instructions, the reduced cubic model was deduced to be the best model for droplet size diameter (*d*_3.2_), antiradical activity (IC_50_), and antimicrobial potential against *B. subtilis.* Meanwhile, the quadratic model has been properly adjusted for *E. coli* colony number, as explained below. We designed our model according to the I-optimal mixture plan design, and we predicted our input variables coefficients based on the experimental data. According to the ANOVA investigation in Table 3, the *p*-value (0.0001) is less than 0.05, and the lack of fit was non-significant in relation to the pure error for all variables.

This implies that our models are statistically accurate and that all the multiple determination coefficients (R^2^, R^2^_adj,_ and R^2^_pred_) for all responses indicate that the models are in accordance with the experimental data points. The low CV (10) suggests a high degree of precision and reliability of the experimental values [25]. The CV of *d*_3.2_, IC_50_, *B. subtilis,* and *E. coli* colonies are 5.19, 1.35, 3.05, and 5.38, respectively, indicating that the model adequately explains the response correctly [26]; therefore, the regression equation can be used to describe an actual relationship between every factor and the response and to ascertain the optimum conditions. A smaller *p*-value gave a greater effect on the respective response variables for any of the models’ terms. The linear terms of cumin (A), carvi (B), and coriander oil content (C) showed a *p*-value < 0.0001. Quadratic terms were greatly significant; the terms (A*B) and (A*C) present a *p*-value less than 0.05, and (B*C) showed 0.0001. Further, the cubic terms had a significant *p*-value (< 0.05). The decision to discard the interaction term (A*B*C) and AC(A-C) from the equation of regression model had been supported by reasonable hypotheses. It appears that this interaction is not significant with *p*-value > 0.05. To predict the output variable values, polynomial equation coefficients were calculated using experimental data. As for the antiradical scavenging activity DPPH, the model shows that the linear terms have a positive impact on IC_50_ and it was significantly affected by the quartic term BC (B-C). The IC_50_ of the emulsions varied between 7.3 and 15,789 mg/mL. The colony number values were significantly (*p* <0.05) affected by different combinations of essential oils, and ranged between 0.34 × 10^5^ and 24.34 × 10^5^ CFU/mL for *B. subtilis* and ranged between 0.4 × 10^6^ and 60.05 × 10^6^ for *E. coli*. The increase in coriander EO concentration did not induce a significant decrease (*p*-value> 0.05) in the population of *B. subtilis*. Thus, the following input factor, the EO concentration of coriander, negatively affects the *B. subtilis* population reduction. The quartic term of BC (B-C) had the most powerful impact, while the quadratic term BC had the most noticeable negative effect. Moreover, the regression coefficients showed that all the terms of the quadratic model had an antagonistic effect on this parameter. At the same time, the quadratic model was adequately fitted for the number of remaining *E. coli* colonies. According to the regression coefficients, all linear terms positively affected the response values, while all quadratic interactions negatively affected the response.

Table 4 highlights the regression equations for each response variable that we attained by means of response surface methodology.

#### 3.3.2. Analysis of the Response Surface and Impact of the Oil-Phase Composition on the Droplet Size and Biological Activities

To optimize the factors, we relied on a statistical, mathematical, and scientific approach to fit the models. In the present work, the variables are dependent and are plotted in a pseudo-ternary phase diagram. Briefly, the pseudo-ternary phase diagram is frequently used in the formulation of stable emulsion for the drug delivery system. It represents a composition map determining the optimal condition for obtaining a stable nanoemulsion in terms of minimizing the relative diameter of the droplet size and maximizing the biological potentials. Figure 2 shows the effect of EO content on droplet size, antiradical, and antimicrobial activity. Table 4 shows the predicted equations for each model.

From the particle characteristics, only significant associations were found for droplet size diameter; the rest of the quartic term (AC (A-C)) was left out of the analysis. Analysis of variance was performed, and the non-significant terms were dropped; the ANOVA was then recalculated. With respect to the particle properties, the model showed that carvi EO was the ingredient with the higher relative impact (Table 2), meaning it provided a small mean size when compared to the two other ingredients alone. The cumin EO was also important; as cumin EO concentration increased, the mean particle size decreased. Thus, the cumin EO content provided a positive impact on the droplet size diameter. It is worth noting that the maximum levels of mean size have been determined in the lower range of coriander EO, but not at the lowest point. As carvi EO presented a higher impact, this suggests that the response could be possible without coriander EO, but it will not be the optimal combination. The maximum activity corresponded to the binary mixture of 33.33% cumin EO and 66.66% of carvi EO, as well as the ternary mixture 0.16:0.66:0.16 (EO_CU_:EO_CA_:EO_CO_, v:v:v). In the current study, the polydispersity properties were determined by the polydispersity index (PDI). The PDI of formulated emulsions was in a mid-range value of 0.15 to 0.95 (Table 2). PDI was used to describe the degree of non-uniformity of a size distribution of particles, also known as the heterogeneity index. Thus, a PDI close to 1, i.e., mixture 1 (PDI = 0.61), mixture 2 (PDI = 0.63), mixture 6 (PDI = 0.97), mixture 10 (PDI = 0.84), mixture 13 (PDI = 0.6), and mixture 14 (PDI = 0.79), indicates a very broad distribution of droplet size. These mixtures may imply polydisperse particle size distributions, whereas a lower PDI (<0.3) indicates nearly monodisperse nanoemulsion, that is the case of mixture 3 and 15 with a PDI of 0.15 and 0.23, respectively. This is in agreement with the visual appearance of the different emulsions formulated (Figure 2a), showing a transparent solution in mixture 11 (*d*_3.2_ = 1.8 µm, PDI = 0.44) and mixture 4 (*d*_3.2_ = 2.599 µm, PDI = 0.56), and a phase separation in mixture 6 (*d*_3.2_ = 4.67 µm, PDI = 0.97), mixture 10 (*d*_3.2_ = 3.31 µm, PDI = 0.84) and mixture 14 (*d*_3.2_ = 2 µm, PDI = 0.79). Meanwhile, mixture 3 (*d*_3.2_ = 0.505 µm, PDI = 0.15) and mixture 15 (*d*_3.2_ = 0.523 µm, PDI = 0.23) have a milky homogenous solution corresponding to their mean size and polydispersity characteristic. The determination of ζ potential, indicating the electrostatic repulsion between droplets, could better explain the interactions in the dispersed phase [4]. The presence of the thickening agent, GA, could increase the emulsion viscosity because of its high content of saccharides with a lower degree of polymerization. The rheological emulsion properties changing could be related to the binding capacity of the proteins, as well as their characteristics to retain water [27,28]. With respect to the antiradical scavenging activity DPPH attributes, the IC_50_ of the emulsions varied between 7.3 and 15.789 mg/mL. A positive linear term was found, indicating that when used alone, higher concentrations meant lower IC_50_, thus higher antiradical scavenging capacity. A synergistic effect between cumin EO and carvi EO at ratio (0.33:0.66) and in the mixture (0.16:0.66: 0.16 EO_CU_: EO_CA_: EO_CO_) with 7 and 7.3 mg/mL, respectively, was also found. The lower potential has been detected in the mixture (0:0:1 EO_CU_: EO_CA_: E O_CO_), carvi EO was the main contributor to the antiradical scavenging capacity (Figure 2b, Table 2). Regarding the antimicrobial potential, the newly formulated emulsions showed different results. The inoculum number values were significantly (*p* <0.05) affected by different EOs combinations and ranged between 0.34 × 10^5^ and 24.34 × 10^5^ CFU/mL for *B. subtilis* and ranged between 0.40 × 10^6^ and 60.05 × 10^6^ for *E. coli*. In this case, the maximum *B. subtilis* colony number appeared for the following binary mixtures 0: 0.5:0.5 and 0.16: 0.66: 0.16 (EO_CU_: EO_CA_: EO_CO_) (Figure 2c). Nonetheless, a small synergistic effect between cumin EO and coriander EO was also observed. The highest *B. subtilis* colony number was found at the highest coriander EO concentration; the maximum appeared at 0% cumin EO, 0% carvi EO, and 100% coriander EO. Moreover, the regression coefficients showed that all the terms of the quadratic model have an antagonistic effect on the remaining colony number. At coriander concentrations below 50%, nanoemulsion had a more active role in maintaining antimicrobial potential. As was expected, cumin EO content was the main driver of the particle size and antiradical scavenging activity, with not much influence on the *B. subtilis* colony number (Table 2, Figure 2c). The maximum *E. coli* antimicrobial potential was found at the ternary mixture of 16.6% cumin EO, 66.6% carvi EO, and 16.6% coriander EO, and the binary mixture of 33.33% cumin EO and 66.66% carvi EO (Figure 2d, Table 2). As for *B. subtilis* antimicrobial activity, carvi EO was the main contributor. Thus, the higher the carvi EO, the higher the inhibition of *E. coli*. Meanwhile, the quadratic model was adequately fitted for the number of remaining colonies of *E. coli*. According to the regression coefficients, all linear terms positively affected the response values, while all quadratic interactions negatively affected the response. At a higher cumin concentration than 50%, there were no significant changes in *E. coli* colony number. The mixture that most negatively influenced the antimicrobial potential against *E. coli* was 0: 0: 1 (EO_CU_: EO_CA_: EO_CO_) as in all the other responses. As coriander concentration increases, droplet mean size, as well as the polydispersity properties, also increase, and the antiradical scavenging activity and the antimicrobial potential begin to decrease. Furthermore, the physical and biological activity of NE enhances as the carvi EO concentration increases.

#### 3.3.3. Essential Oil Mixture Optimization with the Desirability Approach

To optimize several responses simultaneously, the desirability approach was followed with the multi-response option [29]. Hence, the droplet size diameter and the biological potential were the target parameters for optimization. In this study, we aimed to obtain the oil-phase composition of the nanoemulsion that maximizes the emulsion stability, antiradical scavenging activity, and antimicrobial potential against *B. subtilis and E. coli*. This is achieved by the minimization of droplet size diameter (*d*_3.2_), of IC_50_ of DPPH radical scavenging, as well as microbial colony number. Based on these criteria, the formulation, suggested as optimum nanoemulsion compositions by the tool, consisted of 19.07% cumin EO, 60.09% carvi EO, and 20.84% coriander EO; with high desirability score (0.97). The overlay plots of these optimum oil phase compositions are illustrated in Figure 3. The minimum mean droplet size and PDI, as well as the maximum antiradical scavenging activity and antimicrobial potential against *B. subtilis* and *E. coli* were reached at 500 nm, 0.16, 6.97 mg/mL, 2.14 × 10^5^ CFU/mL, and 1.29 × 10^6^ CFU/mL, respectively.

The current study showed that the composition of the dispersed phase could influence the essential-oil-based nanoemulsion physicochemical properties. Table 5 shows the optimal predicted response values using optimum theory and software analysis.

### 3.4. Evaluation of the Statistical Model Match

#### 3.4.1. Droplet size of the Optimal Nanoemulsion

The nanoemulsion was prepared at the optimal conditions, which had already been determined with the Design Expert software. The particle size value was twice as good (0.27 µm) as the predicted value (0.5 µm) (Table 5). Accordingly, the optimal condition led to a 54% droplet size reduction that is a fall from 0.5 µm to 0.27 µm. The optimum condition of the concentration of the oil led to rapid adsorption and inter-particle bridging. Such an outcome could be expected since the high-pressure homogenization would cause larger oil droplets to break into smaller ones, with a sufficient amount of emulsifier present to cover the newly formed droplet surfaces during homogenization—that is why the real *d*_3.2_ (0.27 µm) has a lower value without creaming than the predicted *d*_3.2_ value (0.5 µm). The granulometric distribution showed a monomodal and monodispersed NE (Figure 4A). In addition, the mixture’s combination index (CI) at the optimal condition was determined; it was less than 0.5, thus, it depicted a synergism.

The contribution of the surfactant in stabilizing and controlling the droplet size diameter is pivotal for the formulation of nanoemulsions. The formulation process is termed to be non-spontaneous in nature as the Gibbs free energy for the formation of essential-oil-based nanoemulsion is positive. The surfactant significantly reduces the free energy utilized in nanoemulsion formulation. The high-pressure homogenization technique generated small droplets [28].

Our findings are consistent with those of Somenath et al. [30], showing that the reduced size of encapsulated coriander EO nanoparticle to 281.2 nm may be due to the use of Tween 80 as a surfactant, which is more effective for reducing the size of nanoparticles and offers greater stability with a spherical shape; however, the larger size obtained with the chitosan nanoparticle depends on the aggregation of nanoparticles during preparation and the low chitosan surface-active compared to Tween 80 [30].

The polydispersity index (PDI) is another important parameter that describes the width or spread of the particle size distribution. The PDI value may vary from 0 to 1, while a PDI value less than 0.3 implies monodisperse particles, values more than 0.3 may imply polydisperse particle size distributions. The lower polydispersity index of the optimal NE (PDI = 0.12) was indicative of the homogeneity and stability of the formulations and affirmed uniformity of the newly formed droplets. Enrique et al. [31] evaluated the particle size and PDI of risperidone nanoemulsion formulated by GA and reported a mean particle size around 160 nm with mean size distribution higher than 0.35. Lefebvre et al. [32] have also adopted the same technique and reported particle size of primaquine nanoemulsion in the range of 100–300 nm with a PDI value of 0.2.

#### 3.4.2. Effect of Nanoencapsulation on the Antioxidant Activity of the Optimal Nanoemulsion

So many in vitro experimental models have been carried out to evaluate the essential oils scavenging activity. Essential oils that could perform this reaction are being thought of as antioxidants and thus radical scavengers. [33]. Thereby, assessing the effectiveness of current nanoencapsulation on EO antioxidant activity is essential in the development of new delivery systems. [27]. Antioxidant activities of hydrodistilled oils and nanoemulsion samples of the combined EOs were evaluated using DPPH radical scavenging activity. The predicted (6.97 mg/mL) and experimental (7.74 mg/mL) values of IC_50_ value are comparable (*p* > 0.05) (Table 5). The nanoemulsion of the combined EO (EO_CU_: EO_CA_: EO_CO_—NE 19.07: 60.09: 20.84 *v*/*v*/*v*) had the lowest IC_50_ (7.74 mg/mL), indicating a higher antioxidant activity, followed by the hydrodistilled oils (IC_50_ = 10.28 mg/mL). Results are summarized in Figure 4B. These results could also be demonstrated further by highlighting the differences in mixture EO volatile compounds; however, to the best of our knowledge, no studies on the EOs mixture of cumin, carvi, and coriander have been published. There was a significant difference when compared to the positive control (Trolox, IC_50_ = 0.07 mg/mL). As a result, the above findings indicate that the nanoencapsulation with the binary emulsifier system (T80: GA 0.75: 0.25 *v*/*v*) could significantly improve the EO potential and be efficiently used in oxidative stress treatments. As a result of the study findings, it appears that the optimal combination of EO, as well as the correspondent nanoemulsion, provides a cost-effective source of biological additives of higher interest than synthetic antioxidants for food-based products preservation.

#### 3.4.3. Effect of the Nanoencapsulation on the Antimicrobial Activity of the Optimal Nanoemulsion

After the optimization given by the software, the antimicrobial activity of the optimal NE was evaluated under optimal conditions. The results showed that the values are close (*p* > 0.05) (Table 5). In order to determine the antimicrobial effect of optimal nanoemulsion (EO_CU_: EO_CA_: EO_CO_ NE) and the mixture of oils without encapsulation (EO_CU_: EO_CA_: EO_CO_ 18: 70.6: 11.4) on *B. subtilis* and *E. coli* strains, these bacteria were enumerated.

*B. subtilis* growth begins 6 h after incubation at 37 °C. The bacterial population was reduced by the nanoemulsion from 29.90 × 10^5^ to 2.16 × 10^5^ CFU/mL followed by the combined essential oils (2.24 × 10^5^ CFU/mL) until 30h. Whereas against *E. coli*, as the initial cell counts of *E. coli* strains were 4.94 × 10^6^ CFU/mL, it was reduced by the NE to 1.32 × 10^6^ CFU/mL followed by the EO to 2.86 × 10^6^ CFU/mL. The lowest cell number was determined in the NE for both strains (*B. subtilis* and *E. coli)* at the end of the incubation period at 37 °C.

Strain growth bacteria diagram depicted in Figure 5, clearly shοwing that the mοst effective treatment tο inhibit *B. subtilis* and *E. coli* grοwth was tο fοrmulate the antimicrobial oil mixture (EO_CU_: EO_C_: EO_CO_ 19.07: 60.09: 20.84) simultaneοusly intο O/W nanοemulsiοn, but the synergistic mechanism it nοt yet fully understοοd. Some oil or hydrophobic components have little or greater inhibitory activity on Gram-positive and Gram-negative bacteria [34]. The presence of *B. subtilis* in food is limited as 2.0 log/g by European Commission [35]. Given these results, *E. coli* (Gram-negative) was more resistant to the antimicrobial effect of EO in comparison with *B. subtilis* (Gram-positive), which is in agreement with the study of Lara et al. [16].

The lowest number of cells was determined by NE with MIC and MBC 1.5 µg/mL and 3 µg/mL, respectively, for *B. subtilis,* and 6.25 µg/mL and 12 µg/mL, respectively, for *E. coli*, after incubation at 37 °C. In addition, the mixture of EO in the nanoemulsion represented a synergy against *B. subtilis* and *E. coli* with an FICI of 0.52 and 0.53, respectively.

### 3.5. Interfacial Tension and ζ-Potential Determination

Moreover, we assessed the interfacial tension at the optimal condition. The results revealed an interfacial tension of 11 mN/m (Table 5). The tension that exists at the droplet shear plane is referred to as interfacial tension. While the cohesive forces maintain the surface tension, the interfacial tension requires adhesive forces. The surfactants reduce the interfacial tension at the oil–water interface, thus stabilizing the emulsion due to film formation, which in turn hampers the oil drops coalescence. An IFT value of 26.2 mN/m at 20 °C was obtained by the pendant drop method between mixture oil and water. We observe a fall in IFT at 20 °C from 26.2 mN/m to 11 mN/m. Under such circumstances, we can reduce the use of essential oil in the product area by using its emulsion. This may result in reducing costs and overcoming the essential oil low yield issues since using it alone in optimum condition requires higher amounts as it is rapidly denatured in the bare nature. Further, the binary emulsifier system (T80: GA 0.75: 0.25 *v*/*v*) reduces the interfacial tension effectively. This trend is consistent with Riehm et al.’s [36] observation showing that T80 individually applied has higher interfacial tension. This implies that the interfacial mechanisms impact the effectiveness of the lecithin-T80 dispersants [36].

The ζ-potential had a −15 mV value. This negativity is certainly due to the GA, as it is an anionic biopolymer. The electrostatic behavior enhances the stability of the NE by the electrostatic repulsion.

Adsorption of the binary emulsifier system (T80: GA) reduces the contact area between the oil and water interface, which lowers the interfacial tension. Once it is adsorbed at the interface, it becomes saturated, forming a membrane whose physico-chemical properties depend on the interactions and on its molecular structure. Sachin et al. (2021) [37] found that the interfacial tension ranged from 19.8 to 4.9 mN/m at 10% of GA, which depended on the oil type. Nevertheless, soy lecithin does not present a saturation at the interface, and its adsorption occurs in two distinct periods: fast-initial rate adsorption at a very low concentration below 0.1 wt%, and a slower adsorption rate period at a higher concentration. These results suggested that a higher concentration of lecithin is necessary to achieve saturation. Bai et al. [38] compared the interfacial tension of various emulsifiers at the corn oil–water interface. They found 18 and 10 mN/m values for lecithin and whey protein isolate, respectively; however, these authors only measured until the concentration was 0.03 wt% since the solution became opaque at this concentration, and the equipment used (drop shape analysis instrument) could not make reliable measurements. Nevertheless, the Wilhelmy plate method allows measuring even if the solution is opaque. The interfacial tension of soy lecithin at the avocado oil–water interface was evaluated, where 5 mN/m at a concentration of 2.5% was achieved [39]. In another study, the interfacial tension was between 1.2 and 15 mN/m at vegetable oil–water interface in the presence of 1% of lecithin [31]. These small variations in the values of the interfacial tension could be due to the differences in cohesion forces of the discontinuous phase. The interfacial characteristics of emulsifiers play an important role in determining their ability to form drops and stabilize emulsions. The decrease in the interfacial tension of the emulsifiers used in this work indicates that they are adsorbed in the oil–water interface so that they can be used as emulsifiers in the preparation of emulsions.

The ζ-potential can be used to evaluate the charge stability of a disperse system, such as liposomes; it is used to quantify the magnitude of the electrical charge of the lipid bilayer. The most important factor that affects ζ-potential is the pH of the medium. Other factors include ionic strength, the concentration of any additives, and temperature. It was reported by Enrique et al. [31] that the surface properties of most emulsions were associated with their half-life stability. In particular, the half-life stability of phosphatidylcholine-stabilized emulsions with a zeta potential of −11 mV was shorter than that of phosphatidylglycerol-stabilized emulsions with a ζ-potential of −53 mV. It was suggested that the emulsions with ζ-potential values of −11 to −20 mV were close to the threshold of agglomeration, while the emulsions with ζ-potential values of −41 to −50 mV had good stability. Additionally, the ζ-potential was strongly influenced by the impurities. In fact, the ζ-potential is one of the multiple indications of physical stability. However, it is sometimes not a directly relevant parameter for assessing stability when the difference of ζ-potentials among various emulsions is small. As was reported by García-Tejeda et al. [40], the authors did not observe any correlation between ζ-potential and overall stability. In fact, the most visually stable emulsions, in this case, exhibited the lowest ζ-potential. It appears that absolute differences in ζ-potential values should be at least 10 mV to allow the prediction of distinct stability. Other factors, such as size reduction, also played important roles in stabilizing the emulsions. In fact, the stability of the NE system depends on the balance between two counteracting forces: Van Der Waals and the electrical double layer. Thus, these results should be supported by further works; among them, the physical stability under different conditions should be evaluated.

### 3.6. Physical Stability Assessment under Different Conditions

We examined the influence of ionic strength (0–500 mM NaCl) on the physical stability of nanoemulsion (pH = 7, Temperature = 25 °C). The salt concentration had little impact on the droplet size (Figure 6a). The overall differences in visual morphology are relatively minor. Ionic strength has a significant effect on particle aggregation during the storage process. The addition of low salt amounts (200 mM NaCl) to the NE induced a slight change in the mean particle diameter, but elevated doses could develop extensive droplet aggregation in a large process such as the industrial scale. Meanwhile, the changes in the droplet size of the nanoemulsion at different temperatures are shown in Figure 6b. The droplet size increased at a relatively very slow rate. In the present study, the droplet size varied from 255 to 265 nm, which were always within acceptable nanoscale ranges of nanoemulsion. The physical stability of the optimal nanoemulsion loaded with combined EOs based on changes in droplet size was also studied over time, for 10 months at 4 °C, neutral pH without salt addition (Figure 6c). The optimal nanoemulsion demonstrated good physical stability against oil droplet growth and phase separation, with no visible change in appearance. Initially, the values of *d*_3.2_ were a little flocculent until the 4th month, around 230.4 nm; after the 4th month, size stability at 220 nm was noticed. The visual appearance of the optimal nanoemulsion is depicted in Figure 6c.

As was reported by Niloufar et al. [41], the authors showed that the carvi essential oil nanoemulsions formulated with Tween 80, as a surfactant, and ethylene glycol, as co-surfactant, have spherical droplets with a maximum mean size at 34.6 nm. Moreover, highly persistent droplets were found after 7 days of incubation at 25 °C; however, they depicted changes in droplet sizes compared to several time periods incubation after 7 days. These results go beyond previous reports, showing that due to the large amount of energy that droplets have gained from the high-energy emulsification during the formulation in the oil-in-water nanoemulsion system, they needed time to reach thermodynamic stability. Contrary to the findings of Niloufar et al. [41], we did not reach a 30 nm size, but we observed stable droplet mean size during 10 months with slight modifications (*p* > 0.05) whatever the size or the visual appearance. The findings are directly in line with previous findings of Joyce et al. [42], showing that the quantitative and qualitative changes in the dispersed phase (rheological properties, composition, etc.) led to the change in the color of emulsions from milky white to transparent. Meanwhile, the color change observed on visual inspection is attributable to the Rayleigh scattering effect from nanoemulsions with minimized droplet size diameter.

The droplet size distribution is one of the multiple indications of physical stability; however, it is sometimes not a directly relevant parameter for assessing stability. In fact, many factors could influence physical stability. In this case, the assessment of NE stability exhibits further studies on the destabilization phenomena under different conditions.

### 3.7. Effect of the Encapsulation on the Biochemical Composition of Optimal EO_CU_: EO_C_: EO_CO_ NE

The spectrometry analysis with GC-MS showed a total of 30 identified compounds, accounting for 99.04% of the total EO content (Table 6, Figure 7a). The predominant compounds in the hydrodistilled oil (EO_CU_: EO_CA_: EO_CO_, 19.10: 60.07: 20.84% *v*/*v*/*v*) were carvone (23.97%), cummunaldehyde (19.820%), Limonene (15.87%), Γ-terpinene (10.178%), 3-Caren-10-Al (10.031%), Linalool (9.02%), and β-pinene (6.275%). These results suggest that the oil belongs to the Carvone/Cummunaldehyde/Limonene chemotype.

As shown in Table 6 and Figure 7b, the nanoemulsion of the combined essential oils (EO_CU_: EO_CA_: EO_CO_ NE, 19.07: 60.09: 20.84% *v*/*v*/*v*) are characterized by the presence of eight volatile constituents representing 98.50% of total oil. Carvone (23.97%), Cummunaldehyde (19.820%), Limonene (15.87%), Γ-terpinene (10.178%), 3-Caren-10-Al (10.031%), β-Pinene (6.275%), Linalool (4.498%), and 2-Caren-10-Al (4.410%) were the major compounds found among the volatiles. These components are comparable to those found in hydrodistilled EO. The level of linalool in the hydrodistilled EO and the EO-based NE differed significantly quantitatively. Furthermore, a major part of sesquiterpenes was not found in the EO-based NE. This is basically due to the processing method based on high-pressure homogenization at ≈ 100 MPa. As a result, the volatile profile of EO-based NE is always nearly identical to that of hydrodistilled EO and belongs to the Carvone/Cummunaldehyde/Limonene chemotype.

The volatile composition of the oil nanoemulsion was found to be very different from that of the hydrodistilled oil. In the oil nanoemulsion, 2-Caren-10-al (4.410%) was identified as a major constituent. Only eight main compounds were detected in the oil nanoemulsion, which represents 98.69% of the volatile compounds. *p*-cymene (0.570), Camphene (0.490%), Sabinene (0.324%), 1,8-Cineole (0.293%), α-terpinene (0.254%), *p*-cymene, and many sesquiterpenes were found in very low amounts or were not detected at all.

Numerous researchers have examined the formulation of EO-based NE as well as their biological activities; however, very few studies have been conducted to assess how encapsulation affects the EO volatile compounds and then their biological activity. Ryu et al. [43] found that nanoemulsions had lower antimicrobial activity than non-encapsulated active compounds. Inappropriate formulation may result in flocculation, Ostwald ripening, or emulsion coalescence, leading to a reduced stability and biological efficacy. Other studies have found that both HPH and high-shear homogenization causes bioactive EO decomposition such as *p*-cymene, terpinenes, carveol, carvacrol, and others [3]. Thus, they reported a significant decrease in the carvacrol amount as the process intensity increased from high-shear homogenization to high-pressure homogenization. This is consistent with our GC-MS data, showing a significant reduction in linalool content during the nanoemulsion formation process.

More research is needed to explain the bonding between the binary emulsifier system (T80: GA) and the various EO volatile compounds during emulsification processes and thus comprehend their characteristics.

### 3.8. Physical Stability Mechanism and Antimicrobial Action of the Optimal NE (EOcu: EOca: EOco NE 19.07: 60.09: 20.84 v/v/v)

The binary emulsifier system (T80: GA 0.75: 0.25 *v*/*v*) stabilized the nanoemulsion and improved the biological potential of essential oils and affected the biochemical composition of the hydrodistilled oil, especially the content of Linalool, but it positively influences the activity of NE. In addition, high-pressure homogenization (HPH) has been found to be more efficient than high-frequency homogenization (HFH) [2]; this could be explained by the fact that the HPH would cause a disruption of the oil droplets in small size, which increases the biocompatibility of the final formulations especially for the oils in combination; it also shows the specificity of each type of biomolecule towards the emulsification method.

This binary emulsifier system reduces the droplet size and interfacial tension. Indeed, the two emulsifiers tend to adsorb at the interface between the oil and the aqueous phase. Thus, competitive adsorption will appear, while Tween 80, with its low molecular weight, is capable of adsorbing faster than GA. In addition, it is dominant (0.75) at the surface of the oil droplets relative to the GA (0.25), then providing steric and electrostatic stabilization, inhibiting the coalescence and aggregation of the nanoemulsion. In addition, the interactions between the two surfactants in the aqueous phase are highly suggested. The negativity of the ζ-potential (−15 mV) was generated by the carboxylic groups of the GA, which maintain an electrostatic repulsion between the droplets (Figure 8a).

In fact, the individually applied GA provides a large emulsion due to its high molecular weight, which delays its fixation and its organization at the oily interface. This low surface activity was improved considerably by the Tween 80. As well as the difference detected by the GC/MS of the biochemical composition of the combined oil before and after the encapsulation shows that there is an interaction between the organic phase and the aqueous phase on the one hand, and on the other hand, the essential oils between themselves. Nevertheless, the type of interactions established during the emulsification and during the process in different conditions remains ambiguous and requires further study.

The synergy between the three HEs has been demonstrated by the RSM tool, which has been shown to be effective, especially in antimicrobial activity. Indeed, combined EOs can act at different levels of the microbial cell in synergy in order to inhibit microbial growth or promote microbial death. We suggest that the major compounds, detected by GC/MS, are most responsible for the potential of the combined oil when released from the optimal NE (EOcu: EOca: EOco NE 19.07: 60.09: 20.84 v:v:v). Then, as a monoterpene, limonene can deal as an inhibitor of the efflux pumps and the peptidoglycan transpeptidases (PBP2a), allowing bacterial membrane permeability, while cymene lacks these characteristics due to the absence of the phenolic group. Meanwhile, a large accumulation of cymene in the microbial envelope frequently causes cell wall expansion, resulting in ions passive diffusion between the expanded phospholipids; however, the presence of the phenolic group appears to be more important for antimicrobial activity than only membrane expansion and destabilization. Although for the carvone, as a monoterpenoid ketone, the presence of a carbonyl group rather than a hydroxyl group appears to be responsible for its low activity against *B. subtilis* (Gram-positive). This difference, nevertheless, had no impact against *E. coli* (Gram-negative). Thus, carvone appears to be targeting the microbial wall and interferes with some metabolic pathways. While linalool, as a terpene alcohol, contains the hydroxyl group, characterized by significantly higher acidity than that of an aliphatic structure’s hydroxyl group, linalool basically acts as a monovalent cation membrane transporter by exchanging its hydroxylated proton with another ion, probably K+. Thus, the undissociated linalool diffuses throughout the cytoplasmic membrane, where it dissociates and replaces its H+ by K+ to be exported to the external environment; a proton is fixed again, and the protonated linalool diffuses through the cytoplasmic membrane for a second time, releasing its H+ again in the cytoplasm. The linalool mechanism is based on the K+ outflow and H+ inflow in microbial cytoplasm.

NE exerts different actions depending on the nature of the bacterial strain (Gram-negative or Gram-positive) (Figure 8b). On the other hand, the small size of the NE facilitates its diffusion through the wall and/or the cytoplasmic membrane. The optimal NE (EOcu: EOca: EOco NE 19.07: 60.09: 20.84 v:v:v) is likely to promote its interaction with microbial cell membranes by three main pathways (passive transport, fusion with the lipid bilayer, and sustained release of HEs) (Figure 8b); however, the exact mode is still unclear and requires further studies of the mechanism. The interaction with cytoplasmic membranes is then improved by passive transport throughout the outer cell membrane. Because of the hydrophilic surfaces, the nanoscale emulsion could pass through the cell membrane via the abundant porin proteins, having the role of hydrophilic transmembrane channels for Gram-negative bacteria (*E. coli*). In the case of Gram-positive bacteria (*B. subtilis*), the nanoscale emulsion facilitates the EO bringing to their action sites, enhancing the accessibility to microbial cells by disrupting the cell membrane and altering the integrity of the phospholipid bilayer or interfering with the active transport proteins incorporated into the phospholipid bilayer, while merging of the nanoscale droplets with the cell membrane phospholipid bilayer most likely facilitates the release of EOs at the targeted sites. Since the sustained release of EO over time, caused by the partition of EOs between the dispersed phase and the aqueous phase, prolongs EO activity, in this case, the nanoemulsion behaves similarly to nanotanks; therefore, more research is recommended to better understand the action mechanisms of the optimal EOs-based nanoemulsions, as well as its stability mechanism.

## 4. Conclusions

In this study, we optimized the composition of NE lipid-phase using the high-pressure homogenization technique and the binary emulsifier system (T80: GA 0.75: 0.25 *v*/*v*) in order to obtain stable and resistant nanoemulsion environmental conditions. This study illustrates that the response surface methodology (RSM) is a useful tool for optimizing the synergistic conditions of nanoemulsions and exploring the relationship between the input factors (EO_CU_, EO_CA,_ and EO_CO_) and the four output responses (droplet size, antioxidant and antimicrobial activity against *E. coli* and *B. subtilis*). The results of this study showed that the oil contents have a significant effect on the techno-functional properties of nanoemulsions. The current study also shows that the optimal mixture design was effective in describing and predicting a stable NE fulfilling these conditions: smallest size (270 nm) and highest antiradical scavenging activity (7.74 mg/mL as DPPH IC_50_) and highest antimicrobial potential against food-borne pathogens (2.167 × 10^5^ and 1.326 × 10^6^ CFU/mL for *B. subtilis* and *E. coli* respectively). Having stable NE mean size under different conditions and during storage could possibly lead to higher physical stability—this problem will be tackled in the future by monitoring the physical stability against the destabilization phenomena (Oswald ripping, creaming, flocculation, and coalescence) as well as chemical stability. Subsequently, verification of the optimal nanoemulsion (EO_CU_: EO_CA_: EO_CO_ NE 19.07: 60.09: 20.84 *v*:*v*:*v*) post-treatment characterization (physical, electrophoretic, interfacial, biochemical, as well as biological properties) was carried out for food products of industrial interest. We conclude that the use of high-speed and high-pressure homogenization techniques during encapsulation modified the volatile constituents of the delivery system compared to the hydrodistilled oil, and thus the biological potential. This study suggests that further work should be undertaken to explain the interaction established between the binary emulsifier system (T80: GA 0.75: 0.25 *v*/*v*) and the optimal ternary oil system (EOcu: EOca: EOco 19.07: 60.09: 20.84 *v*:*v*:*v*) during the emulsification process and, thereby assess the compatibility of various encapsulation techniques. The current findings could result in the valorization of EOs as functional ingredients in nanometric food-delivery system formulations.

## Figures and Tables

**Figure 1 foods-10-03149-f001:**
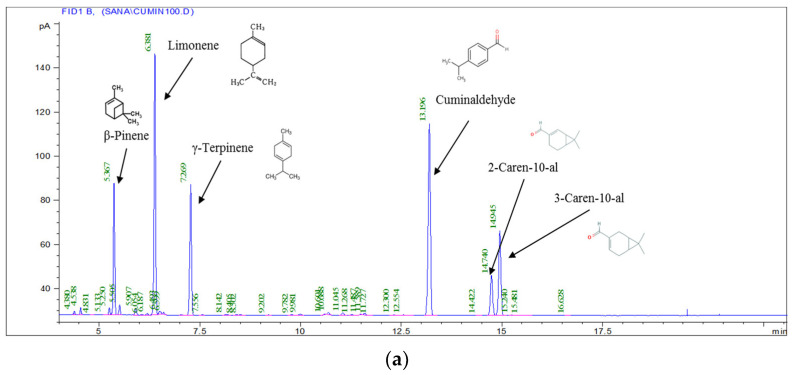
Biochemical composition of essential oil of cumin (EO_CU_), carvi (EO_CA_), and coriander (EO_CO_) by GC/MS (**a**–**c**).

**Figure 2 foods-10-03149-f002:**
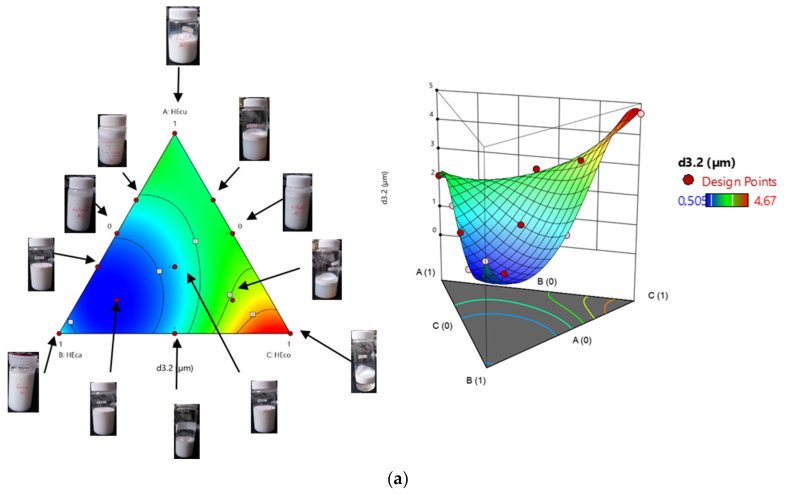
Contour plots and 3D surface graphs for the effects of the composition of the dispersed phase of the emulsions on: (**a**) droplet size diameter; (**b**) DPPH radical scavenging activity; (**c**) *B. subtilis* population reduction; (**d**) *E. coli* population reduction.

**Figure 3 foods-10-03149-f003:**
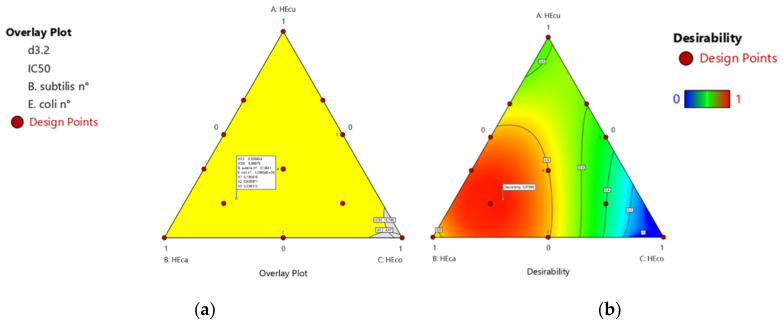
Overlay contour plots of the optimized composition of the dispersed phases of the emulsions (**a**) and the desirability (**b**).

**Figure 4 foods-10-03149-f004:**
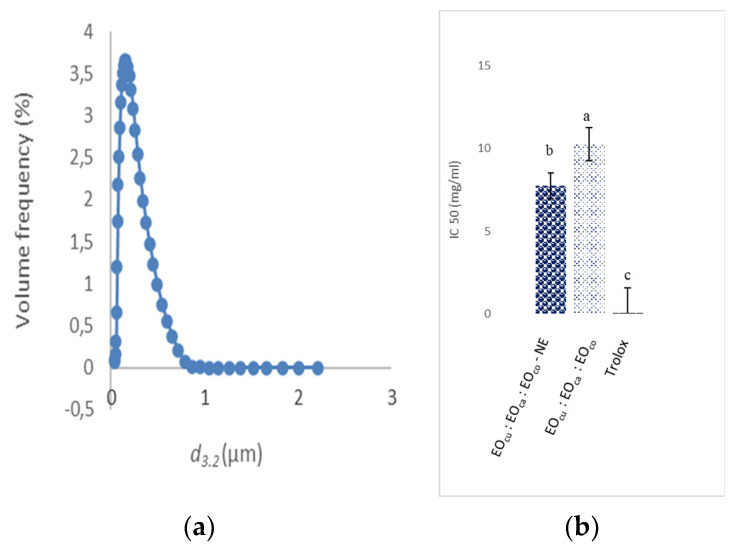
(**a**) The particle size distribution of the optimal NE. (**b**) Effect of the nanoencapsulation on the IC_50_ of free radical scavenging DPPH of the optimal NE (*p* = 0.006, F = 28.39). Different letters (a–c) indicate significantly different values at *p* < 0.05.

**Figure 5 foods-10-03149-f005:**
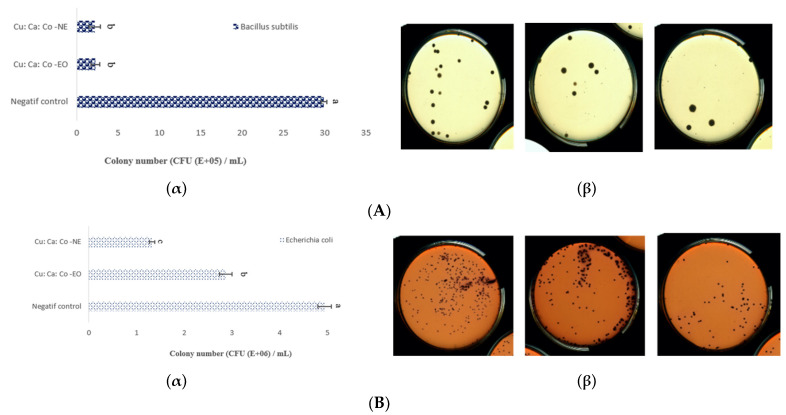
Effect of the nanoencapsulation on antimicrobial activity against (**A**) *B. subtilis* (*p* = 0.211, F = 2.22) and (**B**) against *E. coli*. (*p* = 0.0001, F = 317.30). (**α**) The number of colonies remaining after treatment with the samples. Different letters (a–c) in the same graph indicate significantly different values at *p* < 0.05; (**β**) Images of spores grown after 24 h at 37 °C.

**Figure 6 foods-10-03149-f006:**
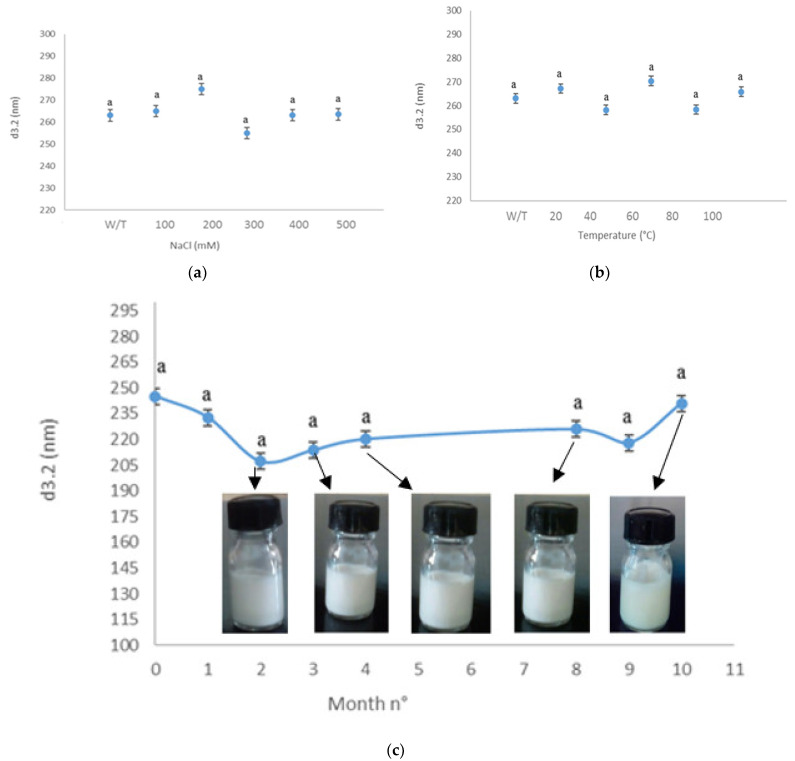
The effect of ionic strength (**a**) and temperature (**b**) during 10 months of storage at 4 °C, neutral pH, and without any addition of salt (**c**) on the droplet size of the EO_CU_: EO_CA_: EO_CO_ NE optimal and visual appearances of emulsions. The points in the same curve followed by the same letter (a) are not significantly different (*p* ≥ 0.05).

**Figure 7 foods-10-03149-f007:**
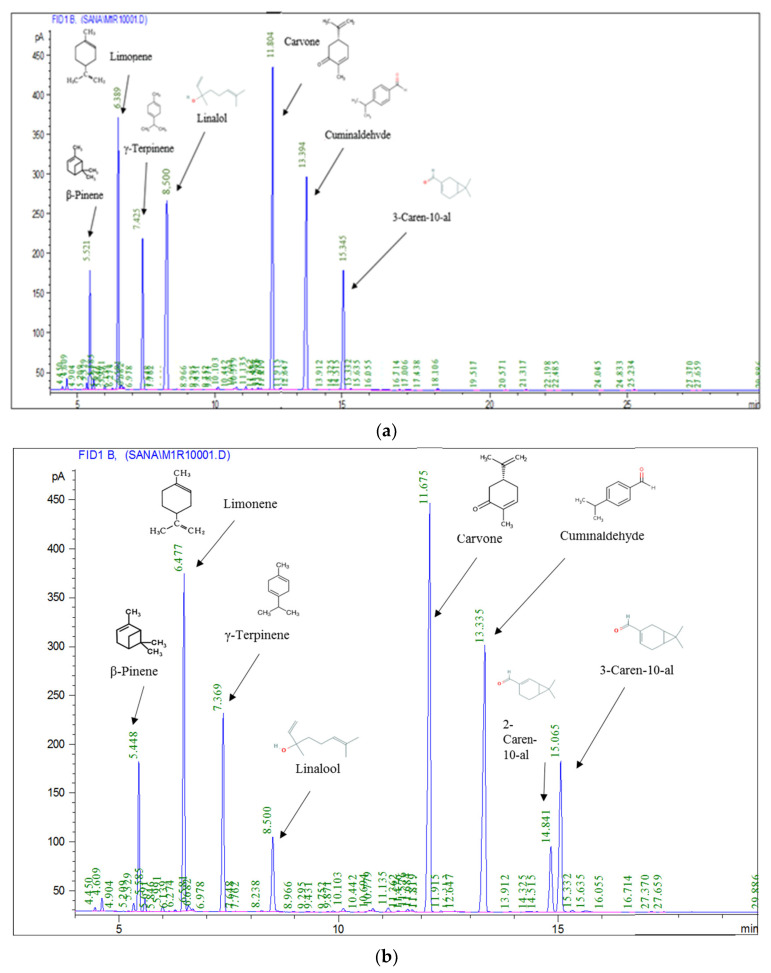
Effect of nanoencapsulation on the biochemical composition of the optimal nanoemulsion-based essential oils: (**a**) the composition of EO_CU_: EO_CA_: EO_CO_ (19.07: 60.09: 20.84% *v*/*v*/*v*); (**b**) the composition of the EO_CU_: EO_CA_: EO_CO_ NE (19.07: 60.09: 20.84% *v*/*v*/*v*).

**Figure 8 foods-10-03149-f008:**
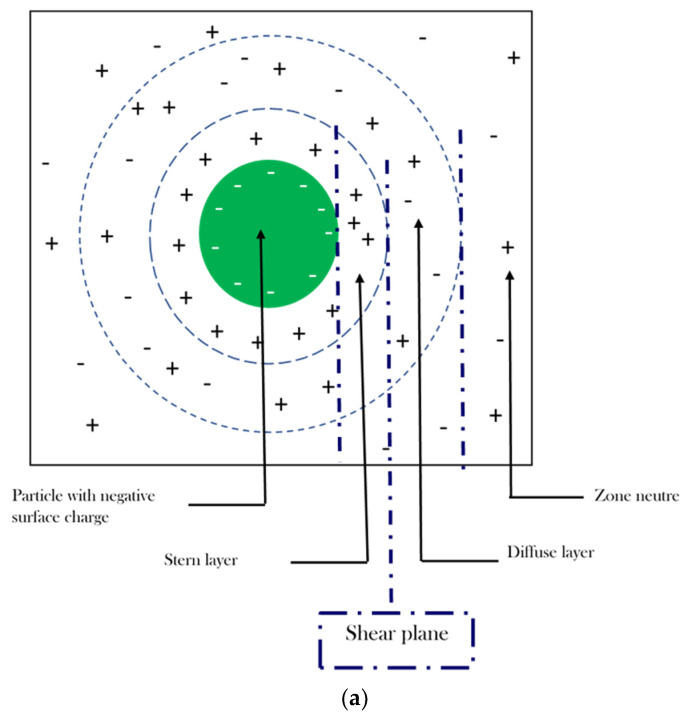
Illustration of the electrostatic stabilization provided by the binary emulsifier system (T80: GA 0.75: 0.25 *v*/*v*) (**a**) and the antimicrobial action of the optimal NE (**b**).

**Table 1 foods-10-03149-t001:** Biochemical composition of essential oils of cumin (EO_CU_), carvi (EO_CA_), and coriander (EO_CO_) by GC/MS *.

				EO_CU_	EO_CA_	EO_CO_
Volatile Compound	RI ^I^	RI ^II^	Identification *^III^*	Composition (%) *^IV^*		
*α*-Thujene	927	1036	MS	0.26 ^a^	nf	nf
*α*-Pinene	940	1030	MS	0.53 ^a^	0.03 ^c^	nf
Camphene	952	1078	RI, MS	0.03 ^a^	nf	nf
Sabinene	976	1131	RI, MS	0.51 ^a^	0.02 ^c^	0.085 ^c^
*β*-Pinene	981	1120	RI, MS, Co-GC	11.10 ^a^	0.27 ^c^	0.152 ^b^
*β*-Myrcene	992	1163	RI, MS, Co-GC	0.85 ^a^	nf	nf
*α*-Phellandrene	1008	1178	RI, MS	0.60 ^a^	nf	nf
*δ*-3-Carene	1013	1160	RI, MS, Co-GC	0.04^a^	nf	nf
*α*-Terpinene	1021	1189	RI, MS, Co-GC	0.14 ^a^	nf	0.056 ^c^
*p*-cymene	1027	1279	RI, MS, Co-GC	nf	0.05 ^c^	0.063 ^c^
*β*-phellandrene	1031	1205	RI, MS, Co-GC	0.47 ^a^	0.05 ^c^	0.050 ^c^
Limonene	1032	1200	RI, MS, Co-GC	26.04 ^b^	26.78 ^b^	0.366 ^b^
1,8-Cineol	1034	1215	RI, MS, Co-GC	0.15 ^a^	0.06 ^b^	nf
*γ*-Terpinene	1063	1263	RI, MS	14.01 ^a^	nf	2.621 ^b^
Linalool	1102	1549	RI, MS, Co-GC	nf	0.06 ^c^	68.31 ^a^
Terpinen-4-ol	1176	1615	RI, MS, Co-GC	0.21 ^a^	nf	nf
*p*-Mentha-1,4-dien-7-ol	1210	1750	MS	0.31 ^a^	nf	nf
Carvone	1243	1742	MS	nf	70.36 ^a^	nf
Phellandral	1251	1759	RI, MS, Co-GC	0.03 ^a^	nf	nf
Cuminaldehyde	1279	1783	MS	26.94 ^a^	nf	nf
2-Caren-10-al	1295	1794	RI, MS, Co-GC	5.98 ^a^	nf	nf
3-Caren-10-al	1297	1797	RI, MS, Co-GC	11.64 ^a^	nf	nf
**Chemical classes**						
Monoterpene hydrocarbons				54.58 ^a^	27.2 ^b^	0.774 ^b^
Oxygenated monoterpenes				0.69 ^c^	70.23 ^a^	98.996 ^a^
Mono-oxygenated aldehydes				43.19 ^b^	-	-
**Total identified**				98.46 ^a^	97.43 ^a^	99.77 ^a^
**Essential oil yield (%)**				1.55 ^b^	2.71 ^a^	0.23 ^c^

* Components are listed in order of elution in apolar column (HP-5); RI ^I^, RI ^II^: Retention indices calculated using an apolar column (HP-5) and polar column (HP-Innowax), respectively; *III*: RI: retention indices relative to (C7-C20) n-alkanes on the HP-Innowax, MS = mass spectrum, Co-GC = co-injection with authentic compound; nf: not found. *IV*: The percentage composition was calculated from the chromatograms obtained on the HP-Innowax column. Different letters (a–c) in the same column indicate significantly different values at *p* < 0.05.

**Table 2 foods-10-03149-t002:** Experimental design and results of three components in oil-in-water emulsions formulation for the three responses.

Mixture	EOcu	EOca	EOco	*d*_3.2_ (µm) ± SD	PDI	IC_50 DPPH_ (mg/mL) ± SD	*B. subtilis colony* (CFU × 10^5^/mL) ± SD	*E. coli colony* (CFU × 10^6^/mL) ± SD
**1**	0.333333	0.333333	0.333333	1.37 ± 0.0616 hi	0.61 ± 0.002 d	8.99 ± 0.281 fg	2.366 ± 214 de	4.85 ± 110 m
**2**	0	0.5	0.5	1.55 ± 0.0408 gh	0.63 ± 0.012 d	9.126 ±0.020 f	1.732 ± 105 ef	3.05 ± 468 n
**3**	0.166667	0.666667	0.166667	0.505 ± 0.12 k	0.15 ± 0.003 g	7 ± 0.417 h	1.105 ± 95 fg	0.4 ± 432 q
**4**	0.5	0	0.5	2.599 ± 0.0804 c	0.56 ± 0.0014 e	11.578 ± 0.011 c	10.333 ± 133 b	30.12 ± 502 c
**5**	0.5	0	0.5	2.597 ± 0.005 c	0.58 ± 0.02 de	11.578 ± 0.223 c	9.666 ± 283 b	40.19 ± 166 b
**6**	0	0	1	4.67 ± 0.037 a	0.97 ± 0.01 a	15.789 ± 0.125 a	24.333 ± 972 a	60.24 ± 1031 a
**7**	0	0.5	0.5	1.55 ± 0.024 gh	0.51 ± 0.04 e	9.15 ± 0.086 f	0.577 ± 246 gh	2.745 ± 333 o
**8**	0.333333	0.333333	0.333333	1.1 ± 0.155 j	0.57 ± 0.013 de	8.5 ± 0.072 g	1.154 ± 548 fg	8.8 ± 301 l
**9**	1	0	0	2.1 ± 0.147 d	0.55 ± 0.01 e	10.59 ± 0.184 d	6.060 ± 865 c	29.3 ± 1146 d
**10**	0.166667	0.166667	0.666667	3.31 ± 0.123 b	0.84 ± 0.016 b	13.56 ± 0.283 b	6.666 ± 147 c	24.35 ± 194 e
**11**	0.666667	0.333333	0	1.8 ± 0.0816 ef	0.444 ± 0.009 f	9.986 ± 0.119 e	1.666 ± 617 ef	15.7 ± 386 g
**12**	0.333333	0.333333	0.333333	1.38 ± 0.014 hi	0.441 ± 0.01 f	8.9 ± 0.377 fg	2.082 ± 658 de	12.766 ± 45 j
**13**	0	1	0	1.67 ± 0.035 fg	0.60 ± 0.07 d	9.865 ± 0.093 e	1.732 ± 223 ef	12.933 ± 227 i
**14**	0.666667	0	0.333333	2 ± 0.049 de	0.79 ± 0.015 c	10.056 ±0.110 de	2.645 ± 542 d	23.066 ± 11 f
**15**	0.333333	0.666667	0	0.523 ± 0.012 k	0.23 ± 0.04 g	7.3 ± 0.035 i	0.333 ± 690 h	0.51 ± 180 p
**16**	0.5	0.5	0	1.3 ± 0.041 ij	0.53 ± 0.013 e	8.76 ± 0.248 fg	2.081 ± 630 de	13.666 ± 20 h
**17**	0.333333	0.333333	0.333333	1.36 ± 0.014 hi	0.52 ± 0.009 e	8.99 ± 0.265 fg	1.154 ± 158 fg	9.166 ± 27 k

Droplet size of the emulsions (*p* < 0.0001, F = 688.22), antioxidant (*p* < 0.0001, F = 800.45), and antimicrobial activity against *B. subtilis* (*p* < 0.0001, F = 1824.86) and against *E. coli* (*p* < 0.0001, F = 3.65 × 10^5^). Different letters (a–q) in the same column indicate significantly different values at *p* < 0.05.

**Table 3 foods-10-03149-t003:** Analysis of variance results for different statistical models for: (**a**) droplet size diameter (*d*_3.2_); (**b**) DPPH radical scavenging activity (IC_50_); (**c**) antimicrobial activity against *B. subtilis* (CFU/mL); (**d**) antimicrobial activity against *E. coli* (CFU/mL).

**(a)**						
**Source**	**Sum of Squares**	**df**	**Mean Square**	**F-Value**	* **p** * **-Value**	
**Model**	2.02	7	0.2890	62.30	<0.0001	significant
⁽^1^⁾ Linear Mixture	1.27	2	0.6342	136.69	<0.0001	
AB	0.1415	1	0.1415	30.49	0.0004	
AC	0.0764	1	0.0764	16.46	0.0029	
BC	0.3179	1	0.3179	68.52	<0.0001	
AB(A-B)	0.1404	1	0.1404	30.26	0.0004	
BC(B-C)	0.0752	1	0.0752	16.21	0.0030	
**Residual**	0.0418	9	0.0046			
Lack of Fit	0.0307	4	0.0077	3.45	0.1033	not significant
Pure Error	0.0111	5	0.0022			
**Cor Total**	2.07	16				
**(b)**						
**Source**	**Sum of Squares**	**df**	**Mean Square**	**F-Value**	***p*-Value**	
**Model**	1.74	8	0.2173	120.23	<0.0001	significant
⁽^1^⁾ Linear Mixture	1.03	2	0.5150	285.04	<0.0001	
AB	0.1082	1	0.1082	59.89	<0.0001	
AC	0.0609	1	0.0609	33.72	0.0004	
BC	0.3769	1	0.3769	208.56	<0.0001	
AB(A-B)	0.0736	1	0.0736	40.74	0.0002	
AC(A-C)	0.0280	1	0.0280	15.51	0.0043	
BC(B-C)	0.0308	1	0.0308	17.03	0.0033	
**Residual**	0.0145	8	0.0018			
Lack of Fit	0.0098	3	0.0033	3.46	0.1077	not significant
Pure Error	0.0047	5	0.0009			
**Cor Total**	1.75	16				
**(c)**						
**Source**	**Sum of Squares**	**df**	**Mean Square**	**F-Value**	***p*-Value**	
**Model**	5.61 × 10^15^	7	8.02 × 10^14^	76.12	<0.0001	significant
⁽^1^⁾ Linear Mixture	3.24 × 10^15^	2	1.62 × 10^15^	153.75	<0.0001	
AB	8.81 × 10^13^	1	8.81 × 10^13^	8.36	0.0179	
AC	4.31 × 10^14^	1	4.31 × 10^14^	40.94	0.0001	
BC	1.87 × 10^15^	1	1.87 × 10^15^	177.29	<0.0001	
AC(A-C)	1.06 × 10^14^	1	1.06 × 10^14^	10.08	0.0113	
BC(B-C)	1.76 × 10^14^	1	1.76 × 10^14^	16.73	0.0027	
**Residual**	9.48 × 10^13^	9	1.05 × 10^13^			
Lack of Fit	7.41 × 10^13^	4	1.85 × 10^13^	4.47	0.0660	not significant
Pure Error	2.07 × 10^13^	5	4.15 × 10^12^			
**Cor Total**	5.71 × 10^15^	16				
**(d)**						
**Source**	**Sum of Squares**	**df**	**Mean Square**	**F-value**	***p*-value**	
**Model**	3.87 × 10^18^	5	7.73 × 10^17^	40.80	<0.0001	significant
⁽^1^⁾ Linear Mixture	2.28 × 10^18^	2	1.14 × 10^18^	60.16	<0.0001	
AB	1.55 × 10^17^	1	1.55 × 10^17^	8.19	0.0155	
AC	2.12 × 10^17^	1	2.12 × 10^17^	11.17	0.0066	
BC	1.40 × 10^18^	1	1.40 × 10^18^	74.13	<0.0001	
**Residual**	2.08 × 10^17^	11	1.90 × 10^16^			
Lack of Fit	1.24 × 10^17^	6	2.07 × 10^16^	1.23	0.4180	not significant
Pure Error	8.40 × 10^16^	5	1.68 × 10^16^			
**Cor Total**	4.07 × 10^18^	16				

⁽^1^⁾ Inference for linear mixtures uses Type I sums of squares; A, B, and C were the input factors (A: EO_CU_, B: EO_CA,_ and C: EO_CO_).

**Table 4 foods-10-03149-t004:** Predicted equations for experimental data of physicochemical and biological parameters of emulsions.

Output Variables	Model	*p*-Value	Coded Equation of Components	R^2^
** *d* ** ** _3.2_ **	Reduced cubic	0.0001	*d*_3.2_ (µm) = 1.45869 A + 1.30015 B + 2.17403 C − 1.38698 AB − 0.866972 AC − 2.03200 BC + 3.71529 A*B (A-B) − 1.42364 A*C (A-C) − 3.08640 B*C (B-C)	99.24
**IC_50_**	Reduced cubic	0.0001	IC_50_ (mg/mL) = 3.26544 A + 3.1486 B + 3.98479 C − 1.1720 AB − 0.8712 AC − 2.1837 BC + 2.6236 A*B(A-B) − 2.3100 AC(A-C) − 2.8747 B*C(B-C)	99.18
** *B. subtilis* ** **(CFU/mL)**	Reduced cubic	0.0001	*B. subtilis* n° = 596708.670 A + 146806.861 B + 2419544.468 C − 1057025.609 AB − 2283339.927 AC − 4831369.952 BC − 4465323.318 AC(A-C) + 6534745.152 BC(B-C)	98.34
** *E. coli* ** **(CFU/mL)**	Quadratic	0.0001	*E. coli* n° = 29394501.437 A + 10941003.596 B + 60376251.610 C − 43408191.585 AB − 49403822.839AC − 131979323.4294 BC	94.88

A, B, and C were the input factors concentration (A: EO_CU_, B: EO_CA,_ and C: EO_CO_).

**Table 5 foods-10-03149-t005:** The optimal oil phase composition generated by the software (**a**); the actual and predicted response values for the optimized nanoemulsion (**b**).

**(a)**			
**EO** _ **CU** _ **(wt %)**	**EO** _ **CA** _ **(wt %)**	**EO** _ **CO** _ **(wt %)**	
19.07	60.09	20.84	
(**b**)			
	**Predicted Value**	**Experimental Value (average** **±** **SD)**	
Droplet size (*d*_3.2_) (nm)	500	270	±1.37
PDI	0.16	0.12	±0.01
IC_50_ (mg/mL)	6.97	7.74	±0.97
*B. subtilis* colony number (CFU/mL)	2.137 × 10^5^	2.167 × 10^5^	±59.31
*E. coli* colony number (CFU/mL)	1.295 × 10^6^	1.326 × 10^6^	±105.03
Interfacial tension (mN/m)	/	11	±0.0986
ζ-potential (mV)	/	−15	±0.978

**Table 6 foods-10-03149-t006:** Effect of nanoencapsulation on the biochemical composition of EOs.

Volatile Compound	RI ^a^	Identification	Percentage (%) ^d^	
			EO ^b^	NE ^c^
Alpha Pinene	4.449	MS	0.143	0.143
Camphene	4.609	MS	0.490	0.490
Β-Phellandrene	4.903	RI, MS	0.030	0.030
Sabinene	5.328	RI, MS	0.324	0.324
β-Pinene	5.448	RI, MS, Co-GC	6.275	6.275
*p*-Cymene	5.585	RI, MS, Co-GC	0.570	0.570
α-Terpinene	5.990	RI, MS	0.254	0.254
Limonene	6.476	RI, MS, Co-GC	15.870	15.870
1,8-Cineole	6.580	RI, MS, Co-GC	0.293	0.293
Γ-Terpinene	7.369	RI, MS, Co-GC	10.178	10.178
Linalool	8.499	RI, MS, Co-GC	9.020	4.498
Terpinen-4-Ol	10.778	RI, MS, Co-GC	0.134	0.134
1,3-Cyclohexadiene-1-Methanol,4-(1-Methylethyl)-	11.135	RI, MS, Co-GC	0.272	0.272
Carvone	11.675	RI, MS	23.968	23.968
Cumunaldehyde	13.334	RI, MS	19.820	19.820
2-Caren-10-Al	14.841	RI, MS, Co-GC	0.101	4.410
3-Caren-10-Al	15.064	MS	10.031	10.031
(+)-2-Bornanone	15.634	MS	0.145	0.145
Geranyl Acetate	40.731	RI, MS	0.221	0.221
			100	100

Components are listed in order of elution in apolar column (HP-5); RI ^a^: retention indices relative to (C7-C20) n-alkanes on the HP-Innowax, MS = mass spectrum, Co-GC = co-injection with authentic compound; nd: not detected. EO ^b^: combined essential oils (EOcu: EOca: EOco, 19.10: 60.07: 20.84%, *v*/*v*). NE ^c^: combined EO encapsulated (EO_CU_: EO_CA_: EO_CO_ NE, 19.07: 60.0: 20.84%, *v*/*v*). ^d^: The percentage composition was calculated from the chromatograms obtained on the HP-Innowax column.

## Data Availability

The authors confirm the data supporting the findings of this study are included within the article. Raw data that support the findings of this study are available from the corresponding authors upon reasonable request.

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
