# Peer review of "Essential-Oil-Loaded Nanoemulsion Lipidic-Phase Optimization and Modeling by Response Surface Methodology (RSM): Enhancement of Their Antimicrobial Potential and Bioavailability in Nanoscale Food Delivery System"

_foods, 2021, doi:10.3390/foods10123149_

Round 1
Reviewer 1 Report
The subject studied in this research is current, novel and provides relevant information within its field. The structure of the manuscript as well as the methodology followed in the study are correct. However, a thorough review of English must be carried out. Some revisions must be carried out and some questions must be answered.
- There are too many abbreviations in the abstract.
- I suggest a review of the English language. For example, and right after starting the manuscript, the sentence that begins on line 47 is strange.
- More current references on the development of nanoemulsions as encapsulation systems should be added in the introduction. Thus, the importance and topicality of the subject is revealed. In fact, there are even studies with optimization by RSM and the use of tween 80.
- Line 161. Actually, there are several high-energy methods for developing nanoemulsions. Therefore, it should be said "an" not "the".
- Was any optimization of the processing method carried out? Sometimes the method is as important as the formulation.
- Why was the sauter diameter chosen instead of the volumetric diameter? Why have no polydispersity parameters been included?
- Only the physical stability was studied from the point of view of the variation of the droplet size? and other phenomena like creaming?
- Examples of droplet size distributions and a more in-depth discussion should be included. Also, some pictures of physical stability over time could be included.
Author Response
Comment : The subject studied in this research is current, novel, and provides relevant information within its field. The structure of the manuscript as well as the methodology followed in the study are correct. However, a thorough review of English must be carried out. Some revisions must be carried out and some questions must be answered.
Authors response: We thank the Reviewer for this assessment of our work.
Comment 1: There are too many abbreviations in the abstract.
Authors response: Thank you for reviewing our manuscript. Corrections have been done and abbreviations in the summary have been corrected.
Comment 2: I suggest a review of the English language. For example, and right after starting the manuscript, the sentence that begins on line 47 is strange.
Authors response: Thank you for asking us to revise the English language. We have made a language editing throughout the manuscript and the sentence on line 47 was replaced by « consumers prefer an eco-friendly product without synthetic chemicals, due to their negative secondary effects. That is why the use of natural aromatic compounds and flavors in different industry products is greatly important ».
Comment 3: More current references on the development of nanoemulsions as encapsulation systems should be added in the introduction. Thus, the importance and topicality of the subject is revealed. In fact, there are even studies with optimization by RSM and the use of tween 80.
Authors response: We agree with the referee and we add references on the development of nanoemulsion in the introduction. Obviously, there are several studies on optimization by RSM, but the key point of the present study is the optimization of the NE lipid phase composition in three essential oils (cumin, caraway, and coriander), which did not still been studied. Likewise, the use of tween 80 combined with gum arabic as an active surface for essential oil NE has not yet been evaluated. We acknowledge that the discussion of related work on nanoemulsion encapsulation systems was incomplete. We have now added several bibliographical references, as mentioned by the reviewer. Paragraphs added in the introduction section in lines (page 2-3 starting line 79 to 105) to reflect this improved discussion on the nanometric delivery system. The references added to the introduction are below.
- Sowmiya, S; Kumaran, R. Is Gum Arabic a Good Emulsifier Due to CH...π Interactions? How Urea Effectively Destabilizes the Hydrophobic CH...π Interactions in the Proteins of Gum Arabic than Amides and GuHCl?. ACS Omega. 2019 4(15): 16418–16428.. https://doi.org/ 10.1021/acsomega.9b01980.
- Cheng, W; Yu, Y; Xiaohong, C; Shangwu, D; Zhong C. Three different types of solubilization of thymol in Tween 80: Micelles, solutions, and emulsions- a mechanism study of micellar solubilization, Journal of Molecular Liquids 2020, 306, 112901. https://doi.org/10.1016/j.molliq.2020.112901
- Tahir, M; Anwaar, A. Tween 80 and Soya-Lecithin-Based Food-Grade Nanoemulsions for the Effective Delivery of Vitamin D. Langmuir 2020, 36, 11, 2886–2892. https://doi.org/10.1021/acs.langmuir.9b03944.
- Espert, M; Salvador, A; Sanz, T. Rheological and microstructural behaviour of xanthan gum and xanthan gum-Tween 80 emulsions during in vitro digestion. Food Hydrocolloids, 2019, 95, 454-461. https://doi.org/10.1016/j.foodhyd.2019.05.004.
- Zhanxiang, W; Da, M; Liang, G; Xuanxuan, L; Yong, W; Octenyl succinate esterified gum arabic stabilized emulsions: Preparation, stability and in vitro gastrointestinal digestion. LWT, 2021, 149, 112022, https://doi.org/10.1016/j.lwt.2021.112022
- Haixia, Z; Qi, F; Di, L; Xing, C; Li, Li. Impact of gum Arabic on the partition and stability of resveratrol in sunflower oil emulsions stabilized by whey protein isolate. Colloids and Surfaces B: Biointerfaces 2019, 181, 749-755. https://doi.org/10.1016/j.colsurfb.2019.06.034.
Comment 4: Line 161. Actually, there are several high-energy methods for developing nanoemulsions. Therefore, it should be said "an" not "the".
Authors response: Thank you for the input. The sentence in section 2.3 « Emulsification method » has been modified as mentioned by the reviewer.
Comment 5: Was any optimization of the processing method carried out ? Sometimes the method is as important as the formulation.
Authors response: We thank the reviewer for reading carefully and going over the materials. The reviewer has commented if there is any optimization in the methodology, as it is very important. All the methodology included in the present work is valid according to previous studies. Although we agree with the reviewer that the high-pressure homogenizer technique at 100 MPa for 4 cycles, was the accepted method in previous work, this has become the standard of essential oil-based nanoemulsion formulation, and so is now mentioned in research reports without further justification (as in the references in cited in our paper). We have already included a citation to the original paper. previous work was carried out to study the effect of HPH pressure and cycle number on the particle size. 100 MPa and 4 cycles were selected as being the best conditions to formulate an essential oil based emulsion with nanoscale diameter and low interfacial tension. If you require further discussion of this method, we will be happy to add a supporting paragraph to the paper. Regarding the NE physical properties determination (droplet size diameter, interfacial tension and z-potential), the references were added in the correspondent section. Whereas, regarding the emulsifier system (Tween 80 : Gum Arabic), yes we have optimized the binary emulsifier system (Tween 80: Gum Arabic). Earlier work have shown that the most stable nanoemulsion was obtained using the binary emulsifier mixture (Tween 80: Gum Arabic 0.75: 0.25 v/v), being utilized as surface-active in a nanometric delivery-system and has proved to be effective not only in the reduction of the interfacial tension (11.3 mN/m) and the droplet size diameter (106 nm) but also in maintaining the biomolecule potential under different conditions of oxidative stress, ionic strength, temperature and during storage, etc.
The chromatographic analysis of essential oil using CPG and GC/MS has been followed by Yakoubi et al,. (2020) and proved to be effective in the determination of volatiles compound of cumin and carvi essential oils. We added the references to the methodology in correspondent section.
Comment 6: Why was the sauter diameter chosen instead of the volumetric diameter? Why have no polydispersity parameters been included?
Authors response: We thank the reviewer for this comment. The particle size distribution can be displayed on a volume, surface area, or number basis. Statistical calculations such as standard deviation and variance are available in either arithmetic or geometric forms. As general rule specifications should be based on the format of the primary result for a given technique. Thus, each technique measures a different property (size) of the particle, and that we may use the data in a number of different ways to get a different mean result (d4.3, d3.2 , etc.). In this sense, the technique used for the droplet size distribution determination is a Laser diffraction apparatus (LS 13320, Beckman-Coulter, Inc., Miami, FL, USA). The Laser diffraction can generate the d4,3 (Volumetric diameter). This is identical to the weight equivalent mean if the density is constant. Furthermore, the most common approach for expressing laser diffraction results is to report also the D10, D50, and D90 values based on a volume distribution. So we believe that each technique is liable to generate a different mean diameter as well as to measure different properties of particles. Thus, the laser diffraction technique generates a volume distribution for the analyzed light energy data. This volume distribution can be converted to any number or length diameter as shown above. Thus, in the current study, we want to compare the essential oil-based emulsions newly formulated, on the basis of surface area because the higher the surface area, the higher the activity of emulsion. That is why the Sauter diameter (d3.2) was chosen instead of the volumetric diameter (d4.3).
Given the interest in the PDI parameter, we have added the PDI values to all emulsions formulated in table 2 and table 5b. Basically, the particle size distribution and polydispersity index (PDI) of oil-in-water nanoemulsion are highly important physical characteristics to be considered when creating food-grade products. Since PDI describes how broad the range of sizes is. For a monomodal dispersion, we would theoretically expect a perfectly exponential decay of the autocorrelation function of the scattered light intensity. Therefore, the PDI measures the homogeneity of nanoparticles, the smaller the PDI, the more homogeneous nanoparticles. Tus, the polydispersity indices affirm uniformity of the droplets formed. The interpretation of the polydispersity index results was quoted in the main text in section 3.3.2. (lines 423 - 454, pages 17-18) and section 3.4.1. (lines 546 - 556, pages 21-22).
Comment 7: Only the physical stability was studied from the point of view of the variation of the droplet size ? and other phenomena like creaming?
Authors response: We thank the reviewer for this comment and we agree. The droplet size distribution is one of multiples indications of physical stability. However, it is sometimes not a directly relevant parameter for assessing stability. In fact, many factors could influence physical stability. In this case, the assessment of NE stability requires further studies on the destabilization phenomena under different conditions. Particle size distribution is sufficient information for the majority of particle characterization applications. If the goal of the measurement is finding small populations of particles than the main distribution, then an investigation of the sensitivity to second distributions should be part of the selection process. Particle shape information may be either desirable or critical depending on the degree to which shape affects product performance. Particle shape influences bulk properties of powders including flow and compaction behavior and the viscosity of suspensions. For food-based product applications, the shape is a critical factor for reflectivity.
The particle size of the dispersed phase is influential to the stability of an emulsion system, which is composed of two immiscible liquids with distinct densities. In this sense, the emulsion with higher resistance and control to creaming should not only contain particles that are small in size but should also be homogenously distributed with relatively low particle size distribution (PSD). Following Stoke’s law, standing the creaming velocity, the stability of emulsion would increase as the droplet size decreases.
ύ Stokes = - 2gr² (ρ2 – ρ 1) / hn1
r is the radius of the particle, g is the acceleration due to gravity, r1 and r2 are the density of two phases, and h is the shear viscosity of the target system.
Therefore, the stability of emulsions and the extent of gravitation separation after a certain time interval could be studied by determining the droplet size under specified temperature and time conditions. In this sense, creaming occurs when the droplet concentration increases in the top layer of the emulsion. Moreover, by monitoring the change in system PSD, one can also identify the incidence of undesirable droplet interaction, which causes the increase in the number of larger particles over time. However, it is still difficult to distinguish flocculation, coalescence, or Ostwald ripening by simply using particle size analysis. Other analytical methods should be carried out when determining the change in sample quality during storage.
We started, in the present study, to assess the physical stability by the changes in mean size, because it is the main parameter that basis of the stability study. Having stable NE with small mean size under different conditions and during storage could possibly leading to higher physical stability. But the problem will be tackled in the future by monitoring the physical stability against the destabilization phenomena (Oswald ripping, creaming, flocculation, and coalescence) as well as chemical stability and sensorial evaluation, as outlined in the section 3.6 (Physical stability assessment under different conditions) and in conclusion / perspectives section of the manuscript.
Comment 8: Examples of droplet size distributions and a more in-depth discussion should be included. Also, some pictures of physical stability over time could be included.
Authors response: We would like to thank the reviewer for pointing out about very interesting works. Rectifications have been done and the currently ongoing discussion focuses more on droplet size distribution. Regarding the pictures of physical stability over time, in figure 6c the visual appearance of optimal nanoemulsion over time was added as the reviewer requested.
We agree with the reviewer regarding the importance of the visual appearance of the nanoemulsion over time because the first impression of the food industry toward any emulsion-based food product is usually formed when viewing its outward appearance. Emulsification gives food products the characteristic appearance and sensory properties that significantly influence the consumer's perception of such products. As human sensation is complex, food processors usually require numerical standards for better quality control. In addition, quality standards become even more important when manufacturers develop new product formulations or adapt innovative processing procedures.
- Somenath, D; Vipin, K. S; Abhishek, K. D; Anand, K. C; Neha, U. Pallavi, S; Sarika S; Nawal, K.D. Encapsulation in chitosan-based nanomatrix as an efficient green technology to boost the antimicrobial, antioxidant and in situ efficacy of Coriandrum sativum essential oil. Int J Biol Macromol. 2019, 133:294-305. https://doi.org/10.1016/j.ijbiomac.2019.04.070.
- Enrique, F.A; Zaira, A. B; Paola, E.S. G; Maribel, J.F. ; César, I.B; Luz A.P. Carotenoid nanoemulsions stabilized by natural emulsifiers: Whey protein, gum Arabic, and soy lecithin. Journalof Food Engineering. 2021. 290, 110208. https://doi.org/10.1016/j.jfoodeng.2020.110208
- Lefebvre, G; Riou, J; Bastiat, G; Roger, E; Frombach, K; Gimel, J-C; Saulnier, P; Calvignac, B. Spontaneous nano-emulsification: Process optimization and modeling for the prediction of the nanoemulsion's size and polydispersity. Int J Pharm 2017 ,534 (1-2), 220-228. https://doi.org/ 10.1016/j.ijpharm.2017.10.017
- Sachin, R. A; Uday, S.A. Microencapsulation of curcumin using coconut milk whey and Gum Arabic. Journal of Food Engineering 2021, 298, 110502. https://doi.org/10.1016/j.jfoodeng.2021.110502.
- Bai, L; Huan, S; Gu, J; McClements, D.J. Fabrication of oil-in-water nanoemulsions by dual-channel microfluidization using natural emulsifiers: saponins, phospholipids, proteins, and polysaccharides. Food Hydrocolloids 2016, 61, 703–711. https://doi.org/10.1016/j.foodhyd.2016.06.0
- Arancibia, C; Riquelme, N; Zúniga, R; Matiacevich, S. Comparing the effectiveness of natural and synthetic emulsifiers on oxidative and physical stability of avocado oil-based nanoemulsions. Innovat. Food Sci. Emerg. Technol. 2017 https://doi. org/10.1016/j.ifset.2017.06.0
- García-Tejeda, Y.V; Leal-Castañeda, E.J; Espinosa-Solisb, V; Barrera-Figueroac, V. Synthesis and characterization of rice starch laurate as food-grade emulsifier for canola oil-in-water emulsions. Carbohydrate Polymers 2018, 194, 177-183. https://doi.org/10.1016/j.carbpol.2018.04.029
- Niloufar, K; Masoud, H.T; Pouran, A;, Samira, Y; Atieh, D. M. Synthesis of Carum Carvi essential oil nanoemulsion, the cytotoxic effect, and expression of caspase 3 gene. J. of food biochem 2019, 43, 8, e12956 https://doi.org/10.1111/jfbc.12956
- Joyce, M.N; Latha, ; Kagitala, A. R; Nagarajan, R. Ultrasonic Nanoemulsification of Cuminum cyminum Essential Oil and Its Applications in Medicine. Int J Nanomedicine 2020, 15, 795-807. https://doi.org/10.2147/IJN.S230893
Thank you very much for the valuable suggestions and comments.

Reviewer 2 Report
Dear Authors,
Your study is very rich in data, but also very poorly presented. Rarely have I seen such poor English quality and excess of spelling mistakes. This is obviously your first draft of this manuscript and you submitted it directly to this journal without a single round of proof-reading. I am not going to list the way too numerous English language and spelling mistakes, because this whole manuscript needs to be rewritten from beginning to end by someone with at least an average proficiency in English language and higher attention to detail. All that aside, I do not believe papers should be rejected solely on the count of writing quality, so I suggest that you carefully revise your manuscript with the help of a professional editor and resubmit.
Incorrect or incomplete information:
-"The GC-MS apparatus was equipped with flame ionization detector". Well, is it a GC-MS or just a GC-FID? Does it have a mass spectrometer coupled with the GC as you claim when you identified some of the essential oil components or not?
-On line 272: what is that "appropriate pH"? Mention the exact pH value so that other scientists can replicate your work accurately.
Author Response
Comment 1: I suggest that you carefully revise your manuscript with the help of a professional editor and resubmit.
Authors response: Thank you for this valuable feedback. We regret that our manuscript needed English editing. The paper has been revised to improve the grammar and readability. If you require further editing, we will be happy to edit it again.
Comment 2: "The GC-MS apparatus was equipped with flame ionization detector". Well, is it a GC-MS or just a GC-FID ? Does it have a mass spectrometer coupled with the GC as you claim when you identified some of the essential oil components or not?
Authors response: Thank you for the comment. The paragraph of section 2.2. « chromatographic analysis » has been corrected. The essential oil components were analyzed by GC-FID followed by GC-MS. The tentative identification of the oil constituents was based on a comparison of their retention indices relative to (C7–C20) n-94 alkanes (Analytical Reagent, LabScan, Ltd., Dublin, Ireland) with those of literature or with 95 those of authentic compounds available in our laboratory. We include more details on the essential composition analysis below :
The composition of obtained essential oils was analyzed using GC and GC-MS analyses. The GC analysis was performed using an Agilent gas chromatograph series 7890-A equipped with a flame ionization detector (FID). The analysis was carried out on fused silica capillary HP-5 MS column (30 m×0.32 mm i.d.; film thickness 0.25 µm). The temperature of the injector and detector was set at 250°C and 280°C, respectively. Nitrogen was used as carrier gas at a flow rate of 1 ml min; the oven temperature program was 60–210°C at the rate of 4°C / min, which was then programmed to 240°C at the rate of 20°C / min, and finally, held isothermally for 8.5 min. The split ratio was 1:50. The GC–MS analysis was carried out by the use of Agilent gas chromatograph equipped with fused silica capillary HP-5MS column (30 m×0.25 mm i.d.; film thickness 0.25 µm) coupled with 5975-C mass spectrometer. The sample volume injected into the capillary column was 0.1 µL pure essential oil in the split mode (1:50). Helium was used as carrier gas with an ionization voltage of 70 eV. The temperature of the ion source and interface was 230°C and 280°C, respectively. Mass range was from 45 to 550 AMU. The oven temperature program was the same as for the GC. The tentative identification of the oil constituents was based on a comparison of their retention indices relative to (C7–C20) n-94 alkanes (Analytical Reagent, LabScan, Ltd., Dublin, Ireland) with those of literature or with 95 those of authentic compounds available in our laboratory.
Comment 3 : On line 272: what is that "appropriate pH"? Mention the exact pH value so that other scientists can replicate your work accurately.
Authors response: We thank the reviewer for this input ; the appropriate pH of the phosphate buffer used for the dilution of the oil-in-water nanoemulsion, in order to determine its electrophoretic properties, is 7 (neutral pH). The electrophoretic properties were determined in neutral condition since the most important factor that affects ζ-potential is the pH of the medium. And our goal in this section was not to assess the change of ζ-potential versus different conditions, but was just the measurement of the electrical charge which could the NE avoid in room temperature, neutral pH. The pH value was added to manuscript in line 272.
Reviewer 3:
Comment 1: Abstract : Few sentences are given as fragments that do not convey the meaning. Please modify.
Authors response: Thank you for giving us the opportunity to improve the abstract. All the fragments have been rectified.
Comment 2 : Line 56: Delete the word “Actually”
Authors response: We agree and removed the word “Actually »
Comment 3: Line 63: Please rephrase as: “Thus, in order to improve the physical, rheological and stability properties of nanoemulsions, the qualitative and quantitative selection of organic and aqueous phase are important factors”
Authors response: We thank the reviewer for the rephrase. The correction has been made as suggested. The sentence is cited from lines 71 to 73.
Comment 4: Line 66: Delete “anyway”.
Authors response: Thank you for this input. The word « anyway » was deleted.
Comment 5: Line 68: The sentence is too long and does not convey the meaning. Please rephrase that sentence.
Authors response: The sentence in line 68 was rephrased by the following sentence (line 65 -68): « However, the study of Yakoubi et al., [3] showed that this technique was unable to generate essential oils-loaded nanoemulsion in the nanometric size. Another technique commonly used is High- Pressure Homogenization (HPH) »
Comment 6: Line 76: Does UHF mean the high-frequency ultrasound (HFU)?
Authors response: Yes. Please forgive us for the typo mistake. This is exactly what we are referring to. We have changed the abbreviation « UHF » by « HFU »
Comment 7: The introduction section is very lengthy and many concepts are not necessary to discuss in detail.
Authors response: Thank you for this comment. We have now rectified the introduction.
Comment 8: As the author used a high-pressure homogenizer for nanoemulsion formulation, why HFU was discussed in the introduction section ?
Authors response: Thank you for allowing us this opportunity to clarify this point. In our previous study, we evaluated the effect of nano-encapsulation on nanoemulsion loaded-EO of cumin and carvi with high-frequency ultrasound (HFU). Thus, in the continuity of our previous research (the reference below), and since the HFU produces an emulsion at the micrometric scale (1.2 µm) influencing the antimicrobial potential, in the current study the emulsification technique has been modified by the high-pressure homogenizer (HPH). We rectified this point in the introduction in lines 64, 65, 66 and 67.
Yakoubi, S; Bourgou, S; Mahfoudhi, N; Hammami, M; Khammassi, S; Horchani-Naifer; K; Msaada, K; Tounsi, M. S. Oil-in-water emulsion formulation of cumin/carvi essential oils combination with enhanced antioxidant and antibacterial potentials. Journal of Essential Oil Research 2020. https://doi.org/10.1080/10412905.2020.1829510
Comment 9: Line 194: Check the spelling of gram-negative and gram-positive bacteria.
Authors response: The spelling of gram-negative and gram-positive has been checked.
Comment 10: What is the purpose of selecting “Gum Arabic” as an emulsifier?
Authors response: We would like to thank the reviewer for pointing out the very interesting point. First of all, Gum Arabic is recognized as an efficient emulsifying agent of the food industry. Furthermore, GA stabilizes the oil-in-water emulsions over a wide range of pH, temperature, and ionic strength. It is considered to be eco- friendly. Secondly, GA is a highly branched, neutral, or slightly acidic, polysaccharidic complex-rich, containing about 2% of the polypeptide. Besides, GA was able to form thick viscoelastic films at the oil-water interface, generating electro-steric stabilization. It should be noted thus that the use of GA as an emulsifier can considerably improve the stability of emulsions to aggregation by reducing the strength of the Van Der Waals interactions between them. The major advantage of GA as an emulsifying agent is that it is a reliable emulsifier as the emulsions stay stable for a very long time. Moreover, GA can inhibit the self-assembly of proteins and promotes their water solubility. As well as the presence of GA, a kind of thickening agent, leads to an increase in the viscosity of NE due to its high saccharide content with a lower degree of polymerization, and proteins are well known for their binding properties and water retention capacities which may be partly responsible for the increase in the viscosity of NE. Therefore, GA significantly improves the rheological properties of NE. Several studies showed the advantages of the use of « Gum Arabic » as an emulsifier, for interest of food-based product application. Finally, GA has been chosen owing to its high solubilization capacity, and low viscosity.
Comment 11: Does only the droplet size measurement is sufficient for measuring the stability? And it may not give reliable results? Quantifying the essential oil content upon storage will provide interesting results.
Author response: We are extremely grateful to Reviewer for pointing out this problem. No, the droplet size measurement only is not sufficient for measuring the stability. The present study should be supported by the determination of other parameters like the creaming index as well as the chemical stability, which will be held in future work. Please see our detailed response to Reviewer #1 above (n° 7). We have also adjusted the text where highlighted in the section 3.6 (Physical stability assessment under different conditions) and in conclusion / perspectives section of the manuscript.
Comment 12: Section 3.6: Authors need to check the figure numbers mentioned in the text and in the figure given. For instance, inside the text, authors mentioned Figure 6a depicted the droplet size during 10 months time. But figure 6a is drawn for the ionic strength.
Authors response: We thank the reviewer for making this point. All the figures numbers mentioned in the text of Section 3.6 have been rectified.
Comment 13: And from the figure, one can say, there is no significant difference in droplet size for 0-500 ionic strength, 20 to 100 C and 0 to 10-month storage. Authors should not conclude the emulsion stability just with the droplet size.
Authors response: We agree with the reviewer. It is important to note that the physical stability is not limited to the droplet size measurement only, other parameters may need to be included. Thus, we cannot conclude on the stability. The stable NE mean size under different conditions and during storage could possibly leading to a physical stability, then it can provide just an idea if the NE has kept its nano-scale diameter. Thus, we cannot decide, further studies are needed. Section 3.6 « Physical stability assessment under different conditions » from line 716 to 719, page 26 has been corrected as suggested by the Reviewer.
Thank you very much for the valuable suggestions and comments.

Reviewer 3 Report
In this manuscript, the authors developed essential oil nanoemulsion and optimized the formulation using RSM technique. Following are my queries and suggestion:
- Abstract: Few sentences are given as fragments that do not convey the meaning. Please modify.
- Line 56: Delete the word “Actually”
- Line 63: Please rephrase as: “Thus, in order to improve the physical, rheological and stability properties of nanoemulsions, the qualitative and quantitative selection of organic and aqueous phase are important factors”
- Line 66: Delete “anyway”.
- Line 68: The sentence is too long and does not convey the meaning. Please rephrase that sentence.
- Line 76: Does UHF mean the high-frequency ultrasound (HFU)?
- The introduction section is very lengthy and many concepts are not necessary to discuss in detail.
- As the author used a high-pressure homogenizer for nanoemulsion formulation, why HFU was discussed in the introduction section?
- Line 194: Check the spelling of gram-negative and gram-positive bacteria.
- What is the purpose of selecting “Gum Arabic” as an emulsifier?
- Does only the droplet size measurement is sufficient for measuring the stability? And it may not give reliable results? Quantifying the essential oil content upon storage will provide interesting results.
- Section 3.6: Authors need to check the figure numbers mentioned in the text and in the figure given. For instance, inside the text, authors mentioned Figure 6a depicted the droplet size during 10 months time. But figure 6a is drawn for the ionic strength.
- And from the figure, one can say, there is no significant difference in droplet size for 0-500 ionic strength, 20 to 100 C and 0 to 10-month storage. Authors should not conclude the emulsion stability just with the droplet size.
Author Response
Comment 1: Abstract : Few sentences are given as fragments that do not convey the meaning. Please modify.
Authors response: Thank you for giving us the opportunity to improve the abstract. All the fragments have been rectified.
Comment 2 : Line 56: Delete the word “Actually”
Authors response: We agree and removed the word “Actually »
Comment 3: Line 63: Please rephrase as: “Thus, in order to improve the physical, rheological and stability properties of nanoemulsions, the qualitative and quantitative selection of organic and aqueous phase are important factors”
Authors response : We thank the reviewer for the rephrase. The correction has been made as suggested. The sentence is cited from lines 71 to 73.
Comment 4: Line 66: Delete “anyway”.
Authors response: Thank you for this input. The word « anyway » was deleted.
Comment 5: Line 68: The sentence is too long and does not convey the meaning. Please rephrase that sentence.
Authors response: The sentence in line 68 was rephrased by the following sentence (line 65 -68): « However, the study of Yakoubi et al., [3] showed that this technique was unable to generate essential oils-loaded nanoemulsion in the nanometric size. Another technique commonly used is High- Pressure Homogenization (HPH) »
Comment 6: Line 76: Does UHF mean the high-frequency ultrasound (HFU)?
Authors response: Yes. Please forgive us for the typo mistake. This is exactly what we are referring to. We have changed the abreviation « UHF » by « HFU »
Comment 7: The introduction section is very lengthy and many concepts are not necessary to discuss in detail.
Authors response: Thank you for this comment. We have now rectified the introduction.
Comment 8: As the author used a high-pressure homogenizer for nanoemulsion formulation, why HFU was discussed in the introduction section ?
Authors reponse: Thank you for allowing us this opportunity to clarify this point. In our previous study, we evaluated the effect of nano-encapsulation on nanoemulsion loaded-EO of cumin and carvi with high-frequency ultrasound (HFU). Thus, in the continuity of our previous research (the reference below), and since the HFU produces an emulsion at the micrometric scale (1.2 µm) influencing the antimicrobial potential, in the current study the emulsification technique has been modified by the high-pressure homogenizer (HPH). We rectified this point in the introduction in lines 64, 65, 66 and 67.
Yakoubi, S; Bourgou, S; Mahfoudhi, N; Hammami, M; Khammassi, S; Horchani-Naifer; K; Msaada, K; Tounsi, M. S. Oil-in-water emulsion formulation of cumin/carvi essential oils combination with enhanced antioxidant and antibacterial potentials. Journal of Essential Oil Research 2020. https://doi.org/10.1080/10412905.2020.1829510
Comment 9: Line 194: Check the spelling of gram-negative and gram-positive bacteria.
Authors response: The spelling of gram-negative and gram-positive has been checked.
Comment 10: What is the purpose of selecting “Gum Arabic” as an emulsifier?
Authors response: We would like to thank the reviewer for pointing out the very interesting point. First of all, Gum Arabic is recognized as an efficient emulsifying agent of the food industry. Furthermore, GA stabilizes the oil-in-water emulsions over a wide range of pH, temperature, and ionic strength. It is considered to be eco- friendly. Secondly, GA is a highly branched, neutral, or slightly acidic, polysaccharidic complex-rich, containing about 2% of the polypeptide. Besides, GA was able to form thick viscoelastic films at the oil-water interface, generating electro-steric stabilization. It should be noted thus that the use of GA as an emulsifier can considerably improve the stability of emulsions to aggregation by reducing the strength of the Van Der Waals interactions between them. The major advantage of GA as an emulsifying agent is that it is a reliable emulsifier as the emulsions stay stable for a very long time. Moreover, GA can inhibit the self-assembly of proteins and promotes their water solubility. As well as the presence of GA, a kind of thickening agent, leads to an increase in the viscosity of NE due to its high saccharide content with a lower degree of polymerization, and proteins are well known for their binding properties and water retention capacities which may be partly responsible for the increase in the viscosity of NE. Therefore, GA significantly improves the rheological properties of NE. Several studies showed the advantages of the use of « Gum Arabic » as an emulsifier, for interest of food-based product application. Finally, GA has been choosen owing to its high solubilization capacity, and low vicosity.
Comment 11: Does only the droplet size measurement is sufficient for measuring the stability? And it may not give reliable results? Quantifying the essential oil content upon storage will provide interesting results.
Author response: We are extremely grateful to Reviewer for pointing out this problem. No, the droplet size measurement only is not sufficient for measuring the stability. The present study should be supported by the determination of other parameters like the creaming index as well as the chemical stability, which will be held in future work. Please see our detailed response to Reviewer #1 above (n° 7). We have also adjusted the text where highlighted in the section 3.6 (Physical stability assessment under different conditions) and in conclusion / perspectives section of the manuscript.
Comment 12: Section 3.6: Authors need to check the figure numbers mentioned in the text and in the figure given. For instance, inside the text, authors mentioned Figure 6a depicted the droplet size during 10 months time. But figure 6a is drawn for the ionic strength.
Authors response: We thank the reviewer for making this point. All the figures numbers mentioned in the text of Section 3.6 have been rectified.
Comment 13: And from the figure, one can say, there is no significant difference in droplet size for 0-500 ionic strength, 20 to 100 C and 0 to 10-month storage. Authors should not conclude the emulsion stability just with the droplet size.
Authors response: We agree with the reviewer. It is important to note that the physical stability is not limited to the droplet size measurement only, other parameters may need to be included. Thus, we cannot conclude on the stability. The stable NE mean size under different conditions and during storage could possibly leading to a physical stability, then it can provide just an idea if the NE has kept its nano-scale diameter. Thus, we cannot decide, further studies are needed. Section 3.6 « Physical stability assessment under different conditions » from line 716 to 719, page 26 has been corrected as suggested by the Reviewer.
Thank you very much for the valuable suggestions and comments.

Round 2
Reviewer 1 Report
Taking into account the answers to the questions as well as the revisions made, the manuscript has been improved well,no further comments.
Reviewer 3 Report
The authors modified the manuscript significantly. However, I noticed many typographical errors. I believe it will be rectified during the production stage.
